# Reassessment of weak parent-of-origin expression bias shows it rarely exists outside of known imprinted regions

Carol A Edwards*, William MD Watkinson, Stephanie B Telerman, Lisa C Hulsmann, Russell S Hamilton, Anne C Ferguson-Smith*

Department of Genetics, University of Cambridge, Cambridge, United Kingdom

**Abstract** In mouse and human, genes subjected to genomic imprinting have been shown to function in development, behavior, and post-natal adaptations. Failure to correctly imprint genes in human is associated with developmental syndromes, adaptive, and metabolic disorders during life as well as numerous forms of cancer. In recent years researchers have turned to RNA-seq technologies applied to reciprocal hybrid strains of mice to identify novel imprinted genes, causing a threefold increase in genes reported as having a parental origin-specific expression bias. The functional relevance of parental origin-specific expression bias is not fully appreciated especially since many are reported with only minimal parental bias (e.g. 51:49). Here, we present an in-depth meta-analysis of previously generated RNA-seq data and show that the methods used to generate and analyze libraries greatly influence the calling of allele-specific expression. Validation experiments show that most novel genes called with parental-origin-specific allelic bias are artefactual, with the mouse strain contributing a larger effect on expression biases than parental origin. Of the weak novel genes that do validate, most are located at the periphery of known imprinted domains, suggesting they may be affected by local allele- and tissue-specific conformation. Together these findings highlight the need for robust tools, definitions, and validation of putative imprinted genes to provide meaningful information within imprinting databases and to understand the functional and mechanistic implications of the process.

**\*For correspondence:**
cae28@cam.ac.uk (CAE);
afsmith@gen.cam.ac.uk (ACF-S)

**Competing interest:** The authors declare that no competing interests exist.

## Editor's evaluation

This manuscript presents a useful meta-analysis of genes with parent-specific expression from mouse-published RNA-seq datasets, focusing on genes with weak allelic bias. A combination of systematic bioinformatic analysis and experimental validation convincingly shows that the number of parentally biased genes has been overestimated and the few novel ones lie at the periphery of known imprinted loci. The work will be of interest to genomicists with an interest in imprinting and its mechanisms.

## Introduction

Genomic imprinting is a mammalian-specific epigenetic process causing some genes to be expressed in a parent-of-origin-specific manner, leading to the functional inequality of parental genomes. In the 35 years since imprinting was first discovered in mammals much has been learned about the mechanisms governing this process and the role epigenetic mechanisms as a whole play in genome function (*Tucci et al., 2019*.) To date, approximately 150 imprinted genes have been identified in mice and humans where they have been shown to have vital roles in development, behavior, and post-natal adaptations (*Tucci et al., 2019*; *Plasschaert and Bartolomei, 2014*; *Cleaton et al., 2014*). Failure to

correctly imprint genes in human is associated with developmental syndromes, adaptive, and metabolic disorders during life as well as numerous forms of cancer (*Ishida and Moore, 2013*; *Uribe-Lewis et al., 2011*).

Canonical imprinting is established during gametogenesis when certain regions of the genome become differentially DNA methylated in the two germlines. Germline differentially methylated regions (gDMRs) associated with imprinted genes differ from others because they are protected from the global demethylation that occurs in the zygote; a process requiring the KRAB-zinc finger proteins Zfp57 and Zfp445 (*Li et al., 2008*; *Takahashi et al., 2019*). Imprinted genes are frequently organized into clusters or pairs in the genome and a single gDMR acts as an imprinting control region (ICR) for the entire domain (*Edwards and Ferguson-Smith, 2007*). Secondary or somatic DMRs are also associated with imprinted genes: these DMRs are established after fertilization, under the control of the ICR, and are generally not bound by Zfp57 and or Zfp445. More recently Inoue et al., identified a germline-derived histone 3 lysine 27 tri-methylation (H3K27me3) mediated mechanism that also can confer parental origin-specific expression. These 'non-canonically' imprinted genes show paternally biased expression in the preimplantation embryo which persists in extra-embryonic tissue but is lost in the embryo proper (*Inoue et al., 2017*).

Advances in RNA-sequencing technology and analysis pipelines have enabled the quantification of allele-specific expression and the identification of imprinted genes in reciprocal hybrids between distantly related strains of mice. Early studies using this method highlighted the challenges of analyzing data derived from reciprocal hybrids (*Babak et al., 2008*; *Wang et al., 2008*; *Gregg et al., 2010b*; *Gregg et al., 2010a*; *Wang et al., 2011b*). For example, in 2010 two related works reported approximately 1300 new imprinted genes in the brain, which would have increased the number of known imprinted genes in the mouse by an order of magnitude (*Gregg et al., 2010b*; *Gregg et al., 2010a*). These datasets were subsequently shown to include multiple false positives emphasizing the need for validation of putative imprinted genes via a complementary method (*DeVeale et al., 2012*).

More recent studies applied improved analytical tools, as well as additional controls, statistical filtering, and validation (*Babak et al., 2015*; *Bonthuis et al., 2015*; *Crowley et al., 2015*; *Perez et al., 2015*; *Andergassen et al., 2017*). The tissues interrogated in these studies vary and each identifies a distinct set of genes with significant parental-origin-specific expression biases raising the possibility that the imprinting status of these new candidates might be cell or tissue-specific and hence overlooked to date. Particular attention has been paid to neurological tissues where a much higher proportion of putative imprinted genes were identified. Interestingly, most of the candidates have significant yet weak allelic biases including those with only 51–60% expression originating from the preferred allele. Here, applying the same analysis tools and validation approach to all datasets, we investigate genes with weak parental origin-specific bias further by focusing on four studies: *Babak et al., 2015* (Dataset A), *Bonthuis et al., 2015* (Dataset B), *Perez et al., 2015* (Dataset C), *Crowley et al., 2015* (Dataset D). It is noteworthy that the different studies have employed different analysis pipelines.

## Results

### Limited overlap between the sets of novel imprinting candidates from RNA-seq

The four studies used for our analysis generated whole transcriptomes from reciprocal hybrid mouse tissues to identify novel imprinted genes. Reciprocal mouse crosses take advantage of sequence-specific polymorphisms to distinguish parental-origin effects from those caused by the genetic background of different mouse strains. A summary comparing the methods used in all four studies is described in the methods section and summarized in *Supplementary file 1a*. We first assessed the overlap between the studies (*Figure 1A and B*). For this, all genes identified were compiled and those with alternative names between studies were merged. This produced a list of 313 genes of which only 36 (11.5%) were identified with allelic biases in all four studies (*Supplementary file 1b*). Genes were then divided into 'Known' (had previously been identified as imprinted or validated using a non-RNA-seq method - *Supplementary file 1b*) or 'Novel' (first identified in these studies or previously un-validated RNA-seq experiments). By our classification, Dataset A identified 99 genes with ASE, of which 26 were novel and 73 were known. Dataset B identified 209 genes, 151 novel and 58 known.

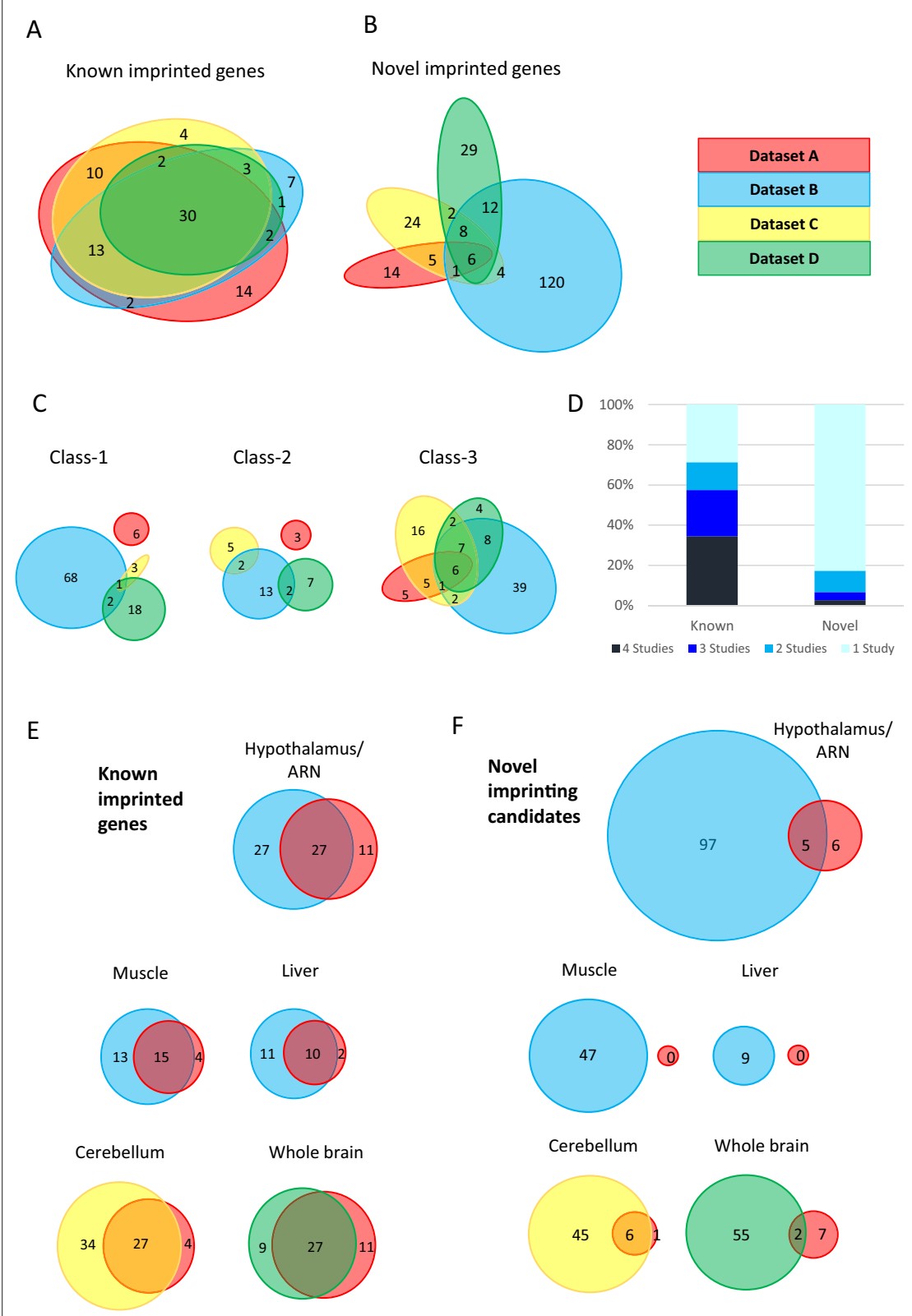

**Figure 1.** limited overlap between novel genes identified in the four studies. (**A–B**) Euler diagrams showing the overlap of (**A**) - known imprinted genes and (**B**) – novel biased genes between the four studies. (**C**) – overlaps between different classes of novel genes. Class-1 = novel singletons >1 Mb from another gene identified in any of the studies. Class-2 = novel clusters where two or more novel genes are within 1 Mb of each other but over 1 Mb from a known imprinted gene. Class-3=novel genes are within 1 Mb of a known imprinted gene. (**D**) - proportion of known and novel genes by number of studies they were identified in. (**E–F**) Venn diagrams showing tissue-specific overlaps in **E** - known imprinted genes and **F** – novel biased genes.

Dataset C identified 112 genes, 51 novel and 61 known, and Dataset D identified 95 genes, 57 novel and 38 known. When the overlap between studies was assessed, 30 out of 87 (35%) known genes were identified in all four studies compared with six out of 226 novel genes (2.7%) (*Figure 1A and B* and *Supplementary file 1c*).

As imprinted genes tend to be clustered within the genome, novel genes were subdivided into three categories. Class-1 are novel singletons that are more than 1 Mb from another gene identified in any of the studies. Class-2 are novel clusters where two or more novel genes are within 1 Mb of each other but over 1 Mb from a known imprinted gene. Finally, Class-3 novel genes are within 1 Mb of a known imprinted gene. The degree of overlap of each of these classes of novel genes varies greatly with a much higher proportion of Class-3 genes overlapping. Indeed, the only six novel genes that came up in all four studies belong to Class-3 (*Figure 1C*) indicating that the boundaries of some imprinting clusters may not have been fully defined previously.

The vast majority (82.7%) of novel parental-origin-specific biased genes were unique to one study, whereas only 28.7% of known genes fell in the same category (*Figure 1D*). The lack of overlap in novel genes suggests that the genes identified in these studies may be subjected to tissue-specific imprinting. Most known genes only identified in one study come from Dataset A. This is not surprising since more tissues were analyzed including four extra-embryonic tissues and 6 of the 14 unique known genes from this study have previously been shown to be specifically imprinted in the placenta or yolk sac (*Wang et al., 2011b*; *Zwart et al., 2001*; *Okae et al., 2012*; *Paulsen et al., 2000*; *Kuzmin et al., 2008*; *Wang et al., 2011a*). To investigate the effect of tissue specificity further, gene sets identified in different studies, but in the same tissue, were compared. Once again minimal overlap was observed between novel genes compared with known genes in the same tissues (*Figure 1E and F*). Thus, despite the careful analytical measures that appear to have been taken in all four studies to reduce false positives, the minimal overlap between the sets of novel genes suggests that some of these may indeed be false positives.

## Reanalysis of data from previous studies indicates differences between studies are due to experimental design and analysis

The lack of overlap between the four studies could be due to the different methods used, and we, therefore, decided to run the sequence data generated from three of the studies (which all used C57BL/6xCastEiJ reciprocal crosses) through the same analysis pipeline and see if the overlap was improved. We utilized the more recently established ISoLDE (Integrative Statistics of alleLe Dependent Expression) package (*Reynès et al., 2020*) that had not been employed in any of the four studies to call allele-specific expression (ASE). ISoLDE uses a nonparametric statistical method to infer ASE in RNA-seq data from reciprocal crosses. It was benchmarked by the authors on six RNA-seq datasets including Datasets B and C used in this study and has been used by others to study imprinted gene expression in the mammary gland (*Xu et al., 2020*).

We chose the hypothalamus, cerebellum, liver, muscle, and whole adult brain from Dataset A, the arcuate nucleus (ARN), the dorsal raphe nucleus (DRN), liver and muscle from Dataset B, and P8 and P60 cerebellum from Dataset C. First, we assessed the overlap between our calls and the calls from the original studies. 204 genes were identified across the five different tissues from Dataset A (*Supplementary file 1d*). 66% of total known genes and 17% of Class-3 genes overlap using their approach and ISoLDE (*Figure 2A*). For these data, ISoLDE identified a high proportion of Class-1 +2 genes (65.2%). No overlap is observed in Class-1 +2 genes as none were identified in the original study. This high number of novel genes called by ISoLDE is most likely due to the absence of biological replicates in Dataset A as other datasets without biological replicates also have increased novel calls (data not shown).

For Datasets B and C, this identified 86 and 63 total ASE genes, respectively (*Supplementary file 1d*). For Dataset B, 49 known genes overlap which is 73% of all genes identified using both methods (*Figure 2B*). This is consistent across all four tissues analyzed with between 50–71% of known imprinted genes overlapping in the tissues (*Supplementary file 1e*). Fewer novel genes show overlap between the two methods: 11% of Class-3 genes and 1% of Class-1 +2 overlap (*Figure 2B* and *Supplementary file 1d*). A similar pattern is observed in the Dataset C with 67% of known genes overlapping, 14% of Class-3, and 8% of Class-1 +2 (*Figure 2C*). For both datasets, fewer novel genes were identified using the common pipeline than in the original studies indicating that the individual methods used to

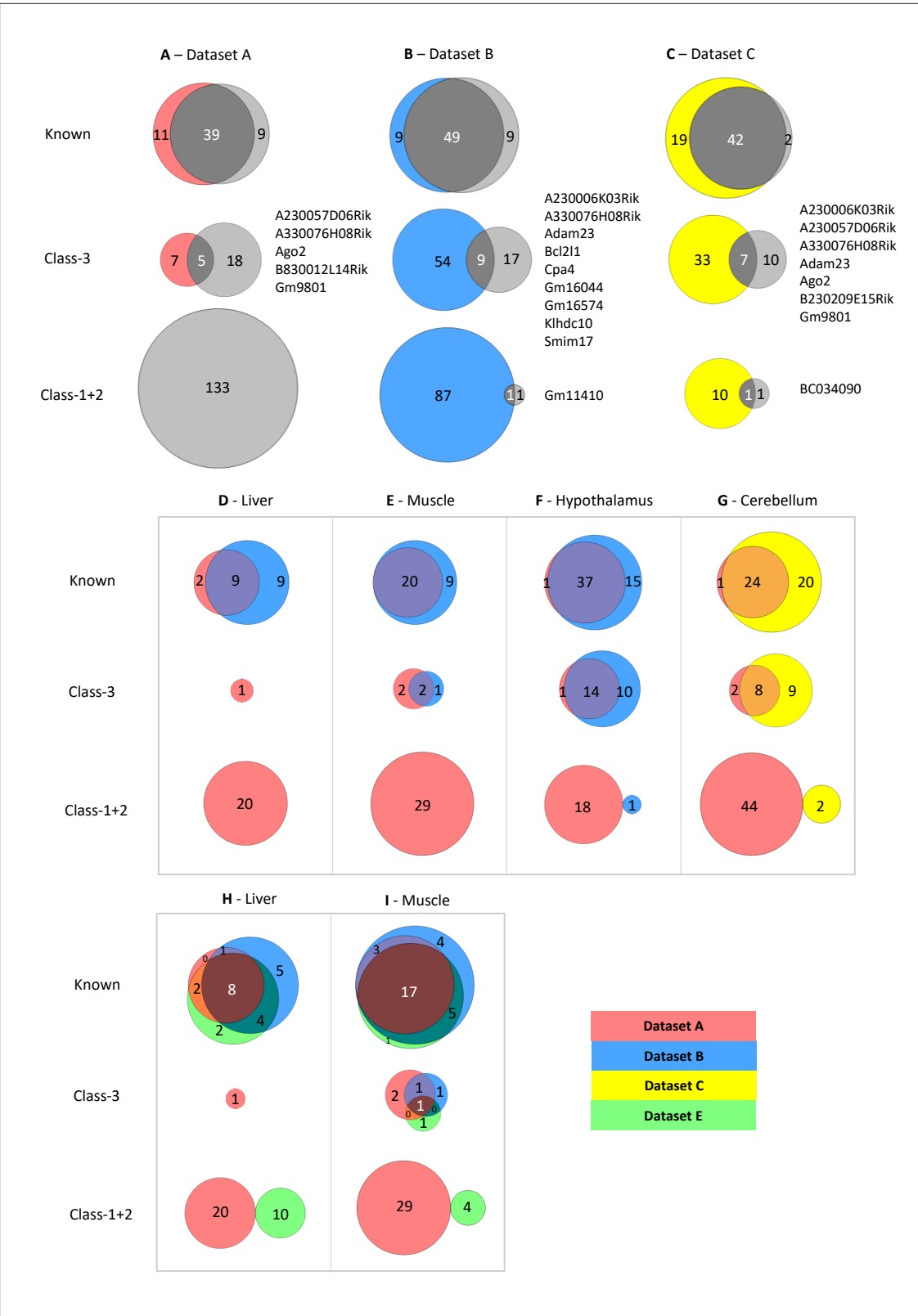

**Figure 2.** Limited overlap between novel genes called by different analysis pipelines. (**A–C**) Venn diagrams showing the overlap between allelic biased genes called by our pipeline with the ISoLDE package (gray circles) versus the original studies. **A** – Dataset A, **B** - Dataset B, **C** - Dataset C. Overlapping novel genes are listed to the right. Dataset A only genes called in the hypothalamus, cerebellum, liver, muscle, and whole adult brain were analyzed. (**D–G**) Venn diagrams showing the overlap between allelic biased genes called by our pipeline from sequence data generated from the same tissue

*Figure 2 continued on next page*

Figure 2 continued

by different studies: liver (**D**), muscle (**E**), hypothalamus (**F**) from Dataset A and Dataset B and cerebellum (**G**) from Dataset A and Dataset C. (**H–I**) Venn diagrams of three-way overlap between allelic biased genes called by our pipeline from sequence data from the liver (**H**) and muscle (**I**) generated by Dataset A, Dataset B, and Dataset E.

The online version of this article includes the following figure supplement(s) for figure 2:

**Figure supplement 1.** Expression levels of genes called as biased in at least one of the original studies.

**Figure supplement 2.** Overlapping underdetermined genes called between Dataset B (**A**) and Dataset C (**B**) and ISoLDE pipeline.

identify allele-specific genes in the four studies greatly influence the calling of novel imprinted genes. Only one Class-1 +2 gene was found in each study: *Gm11410* in Dataset B and *BC034090* in Dataset C. This is a much lower proportion of Class-1 +2 genes than was called in Dataset A (2.3% and 3.2% in datasets B and C, respectively vs. 65.2%) and is most likely due to the higher number of biological replicates sampled in Datasets B and C.

To gain further insight into how methodology affects calling, datasets derived from the same tissue from different studies, but analyzed using our pipeline, were compared. We were able to compare data for postnatal liver, skeletal muscle, and hypothalamus (ARN) between Datasets A and B and for cerebellum between the Datasets A and C. Nine out of 20 known imprinted genes identified in the two liver datasets overlapped (45% - *Figure 2D*), 20 of 29 in muscle (69% - *Figure 2E*), 37 of 53 in the hypothalamus (70% - *Figure 2F*), and 24 of 49 in the cerebellum (49% - *Figure 2G*). Fewer overlaps were found for Class-3 genes: Two out of 5 genes overlapped between the muscle datasets (40% - *Figure 2E*), 14 of 36 in the hypothalamus (39% - *Figure 2F*), and 8 of 24 in the cerebellum (33% - *Figure 2G*). Only one gene was identified in the liver datasets (*Figure 2D*). No overlapping Class-1 +2 genes were identified in any tissue (*Figure 2D–G*).

RNA-seq technology involves PCR which can increase biases in lowly expressed genes through the random amplification of a small pool of molecules. To see if the lack of reproducibility was due to the weakly biased genes being poorly expressed in the original datasets, the expression levels for genes identified as biased in the original studies were compared (*Supplementary file 1f* and *Figure 2— figure supplement 1*). Weakly expressed genes were not overrepresented in the Class-1 and 2 genes. Indeed, all four classes of genes have similar distributions of expression levels, suggesting weak expression is not the only factor affecting the lack of overlap between the two pipelines.

To see if a particular dataset caused the lack of overlap, we next incorporated data from another study (*Andergassen et al., 2017* – Dataset E) that investigated imprinting across multiple tissues and time points in CastEiJ and FVB reciprocal crosses (*Andergassen et al., 2017*). Liver and muscle data were run through our pipeline (*Supplementary file 1d*). The three-way comparison shows a high overlap between the known genes identified in all three datasets (36–57%), variable overlap in Class-3 genes (0–17%) and no overlap between Class-1 +2 called genes (*Figure 2H,I*). Together, these data show that the methods used to generate libraries, the number of biological replicates included, and the methods used to call ASE all greatly influence the genes called.

We next investigated whether strain biases influenced novel gene calling and the impact of this on calling parental-origin-specific expression bias. The overlap between strain specific genes called by ISoLDE and novel genes called in Datasets B and C was compared. Twenty-five genes called as ASE in Dataset B are called as strain-specific by ISoLDE of which 20 are Class-1 +2 (*Supplementary file 1g*) compared with only one overlapping parent-of-origin called gene, suggesting they were miscalled as imprinted in the original study. Five overlapping strain-specific genes were called in the Dataset C: three of them in known imprinted regions and two known imprinted genes – highlighting that strain-specific expression can also act on imprinted genes and needs to be considered when calling allele-specific expression.

## Weakly biased Class-3 genes are peripherally located and preferentially expressed from the chromosome carrying the germline methylation mark

One of the most surprising findings of the cited studies is that many novel imprinting candidates show only a weak bias in the expression of the parental alleles, in some cases only slightly different from an unbiased 50:50 expression ratio. Perez and colleagues grouped their genes according to the

percentage expression from the preferred allele (*Perez et al., 2015*) and we used and extended this grouping to the gene sets from the other studies (*Figure 3A* and *Supplementary file 1b*). Overall, among the 311 known and novel candidate genes for which bias data was available, we observed a bimodal distribution with 169 genes in the 50–60% bias group and 95 genes in the 90–100% group (*Figure 3A*). This bimodal distribution had already been described by Perez and colleagues (*Perez et al., 2015*). Notably, two-thirds of known imprinted genes (77.3%) were in the 90–100% group while only 12.1% of novel candidates show this high expression bias. Most novel genes (73.1%) were found to have only weak biases with 50–60% expression coming from the preferred allele. 88.7% of Class-1 novel singletons and 83.9% of Class-2 genes in novel clusters fall within this weakly biased group (*Figure 3A*). Interestingly, Class-3 genes, close to known imprinted regions, also show a bimodal distribution: 53.7% fall in the 50–60% group and 23.2% in the 90–100% group but, only nine genes (9.5%) show a 70–90% expression bias. The presence of Class-3 genes displaying strong imprinted expression again indicates that the full extent of imprinted expression at some imprinted clusters has not been fully established.

Next, we compared the direction of the parental bias among our four classes of genes. Both Class-1 and 2 genes had a relatively even split between preferentially paternal or maternal expression in both weak and strongly bias genes (*Figure 3B*). Interestingly, although the known imprinted genes consist of about equal proportions of maternally and paternally expressed genes (42 paternal, 33 maternal, 13 genes with both directions depending on tissue), paternally expressed genes tend to have higher expression biases (37 have 90–100% bias), while all eleven known genes with a low expression bias (50–70%) are maternally expressed (*Figure 3B*). Novel genes in known clusters (Class-3) follow the same trend of parental bias and direction (*Figure 3B*) with 70.3% of weakly biased genes preferentially expressed from the maternal chromosome and 64.5% of highly biased genes expressed from the paternally inherited chromosome. Most imprinted regions are controlled by maternal, allele-specific methylation at the ICR acting as a repressed promoter for maternally repressed imprinted alleles. As a consequence of this direct repression, these genes show very strong paternal expression bias whereas paternally repressed genes within these same regions tend to rely on more indirect repression mechanisms such as transcription of a long ncRNA and differential histone modifications (*Fitzpatrick et al., 2002*; *Pandey et al., 2008*). To see if the direction of ICR methylation is causing the trend towards strong paternal and weak maternal biases, we categorized the genes in regions with known ICRs according to whether the preferentially expressed allele was on the chromosome with a methylated (Meth-ICR) or unmethylated ICR (Un-ICR) regardless of parental origin (*Figure 3C*). The direction of methylation does influence the bias as weakly biased genes tend to be preferentially expressed from the chromosome with the Meth-ICR (81% of Class-3 and 100% known <70% biased) whereas strongly biased genes tend to be preferentially expressed from the chromosome with the Un-ICR (80% of Class-3 and 54% known >70% biased -*Figure 3C*).

To investigate this trend further, we assessed the DNA methylation and H3K27me3 at the promoter regions of all the Class-3 and known genes at canonically imprinted clusters using previously published embryonic and postnatal data (*Lister et al., 2013*; *Gorkin et al., 2017*; *Shen et al., 2012*; *Figure 3D* and *Supplementary file 1h*). Strongly biased known genes on the Un-ICR chromosome were associated with promoter methylation suggesting direct regulation by DNA methylation, whilst strongly biased known genes on the Meth-ICR chromosome tend to have unmethylated promoters but higher levels of H3K27me3 indicating repression of genes on the Un-ICR chromosome is controlled by differential histone modifications rather than differential promoter methylation. Interestingly, strongly biased Class-3 genes on the Un-ICR chromosome follow the same trend as the known genes and have methylated promoters indicating a shared mechanism, whereas strongly biased genes preferentially expressed from the Meth-ICR show low promoter methylation and H3K27me3 occupancy. Weakly biased genes tend to be associated with unmethylated promoters and low H3K27me3 regardless of which allele is preferentially expressed. This is unsurprising as the biases are so weak, however, the trend for low biased genes to be preferentially expressed from the Meth-ICR allele implies the genic environment is more repressive on the Un-ICR chromosome perhaps due to ectopic spreading of repressive marks in some cells.

Next, Class-3 genes were classified by whether they were flanked by previously known imprinted genes or were peripherally located within known imprinted clusters. Weakly biased novel genes are predominantly found at the periphery of annotated imprinted domains: 83% of peripheral Class-3

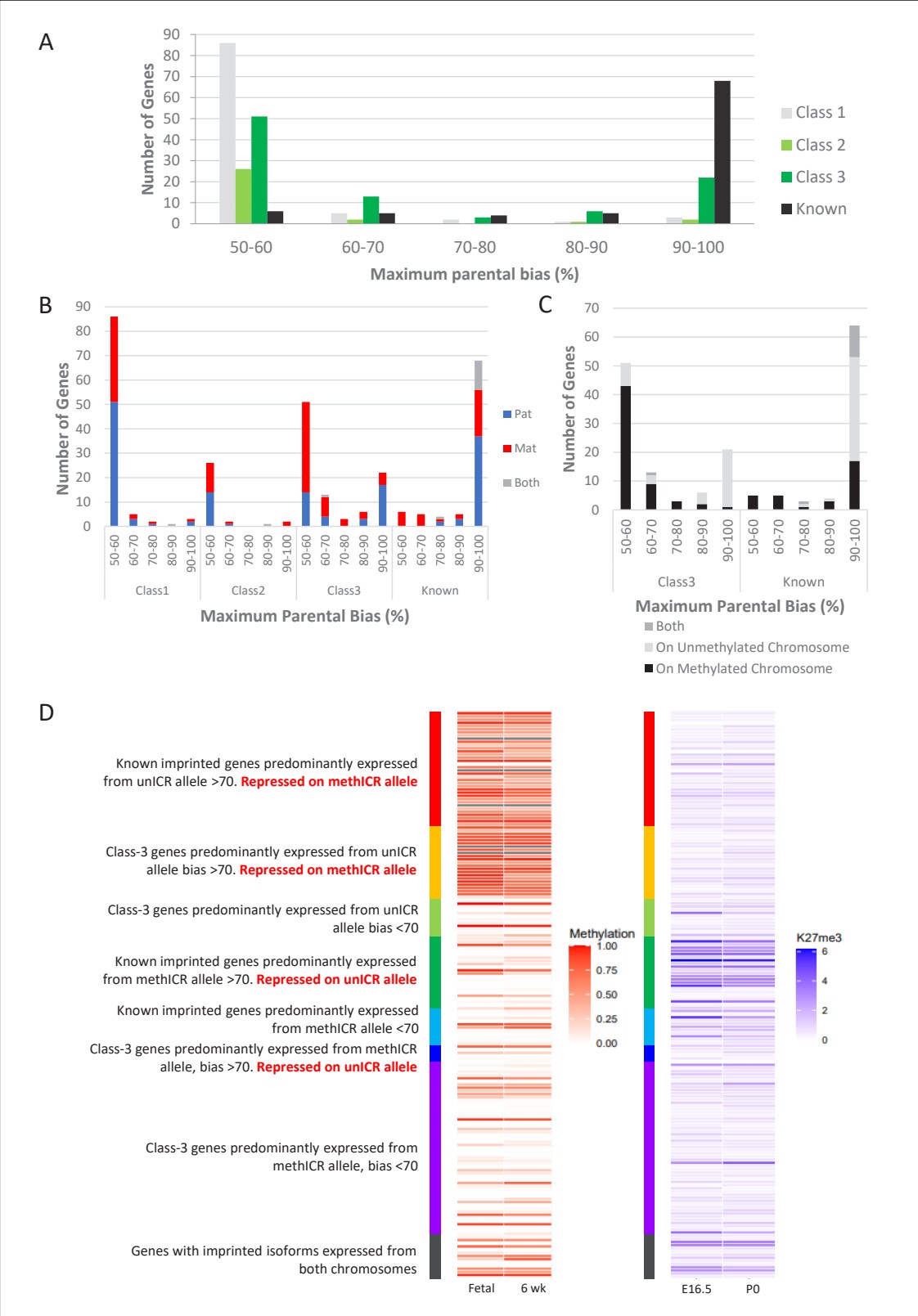

**Figure 3.** Weakly biased Class-3 genes are preferentially expressed from the chromosome carrying the germline methylation mark. (**A**) Distribution of biased genes by maximum reported bias. Class-1 = novel singletons >1 Mb from another gene identified in any of the studies. Class-2 = novel clusters where two or more novel genes within 1 Mb of each other but over 1 Mb from a known imprinted gene. Class-3 = novel genes are within 1 Mb of a known imprinted gene. (**B**) – Distribution of biased genes by preferential parental chromosome and maximum reported bias. Genes preferentially

*Figure 3 continued on next page*

Figure 3 continued

expressed from the maternal chromosome are shown in red, paternal chromosome in blue, and preferentially expressed from both chromosomes in a tissue-specific manner in gray. (**C**) - Distribution of known and Class-3 genes by the methylation status of the imprinting control region (ICR) on the preferentially expressed allele. Genes preferentially expressed from the Meth-ICR chromosome are shown in black, Un-ICR chromosome in pale-gray, and preferentially expressed from both chromosomes in a tissue-specific manner in dark-gray. (**D**) – Heap maps of DNA methylation (fetal and six week male frontal cortex *Shen et al., 2012*; *Sloan et al., 2016*) and Histone H3 lysine 27 trimethylation (E16 and P0 forebrain *Gorkin et al., 2017*; *Shen et al., 2012*) over the promoters of known and Class-3 novel genes. Promoters are defined as 500 bp upstream of the transcription factor binding site. Genes are sorted by maximum reported bias and methylation status of the ICR on the preferentially expressed allele. Source data for the figure can be found in *Supplementary file 1h*.

genes (60/72) have a bias below the 70:30 canonical imprinting threshold set by *Andergassen et al., 2017* (*Figure 4A*). Conversely, highly biased Class-3 genes tend to be flanked by known imprinted genes: only 13% (3/23) of novel genes flanked by known imprinted genes have a ratio below 70:30 (*Figure 4A*). Most of the 20 flanked, highly biased Class-3 genes belong to just two imprinted domains. Fourteen map between *Ndn* and *Snrpn* on chromosome 7 (*Figure 4C*) and five map downstream of *Meg3* on chromosome 12 (*Figure 4D*). In both cases, these highly biased genes follow the direction of imprinting of the long non-coding gene in the region that is repressed on the meth-ICR chromosome suggesting that these may be poorly annotated RNAs arising from known poly-cistronic imprinted transcripts.

We next integrated the position and strength of bias with the ICR status on the chromosome preferentially expressing the gene and found over half of all Class-3 genes are weakly biased, at the edge of the known cluster and preferentially expressed from the chromosome that carries the methylated copy of the ICR (*Figure 4B*). This suggests that secondary repressive mechanisms acting at imprinted genes on the unmethylated chromosome are exerting a small effect on the expression levels of a gene at the periphery of the cluster. Perez et al., previously showed that the degree of bias is reduced as a function of distance from the most strong bias gene in the cluster indicating that the influence of ICRs diminishes over distance. To test whether the ICRs control these weak biases we used the *Dlk1/Dio3* region as a model since it has strongly biased Class-3 genes in the centre of the cluster and nine weakly biased genes were called at the periphery (*Figure 4D*). Utilizing a previously described mouse model with a deletion of its ICR (IG-DMR) that causes a maternal-to-paternal epigenotype switch when maternally inherited (*Lin et al., 2003*), we looked to see if the biased expression of peripheral genes was lost when the ICR was removed. Female mice heterozygous for the deletion were crossed with castaneus males to allow allele-specific expression to be determined by pyrosequencing. *Dync1h1* which is located on the distal side of the region and was reported to be paternally biased in two of the original studies shows fully biallelic expression in E15.5 brain from maternal heterozygotes and wildtype littermates (*Figure 4E*). Conversely, two genes proximal to the defined cluster, *Wars*, and *Wdr25*, both show a weak paternal expression bias in wildtype brains that is significantly reduced in mice with the IG-DMR deletion (*Figure 4E*). This indicates that the weak biases seen at the edges of imprinted clusters are regulated by the ICRs and may be innocent bystanders of the different epigenetic environments established by the ICR on each chromosome.

### Weakly biased genes close to known imprinted domains are more likely to experimentally validate than novel genes elsewhere in the genome

The lack of overlap between studies in the same tissue (*Figure 1F* and *Figure 2D–I*) implies a high number of false positives. To test if these parent-of-origin biases are real we performed quantitative RT-PCR and pyrosequencing validation on eight Class-1, 15 Class-2 genes (representing five novel clusters), and 20 Class-3 genes identified close to seven known imprinted domains. These genes included those that overlapped between the original studies, and those called as ASE or 'Undetermined' by ISoLDE that overlapped with an original study (*Figure 2A–C*, *Figure 2—figure supplement 2* and *Supplementary file 1c and i*). Six known imprinted genes were tested as controls: the robustly imprinted *Dlk1* and *Peg3*, the more moderately biased *Mcts2* and *Herc3*, and the extra-embryonic tissue-specific imprinted genes *Gab1* and *Ampd3* (*Okae et al., 2012*; *Schulz et al., 2006*). Thirteen tissues were analyzed from three different timepoints e16.5, P7, and P60. Expression was called as biallelic if the mean of the paternal expression from both the C57BL/6 × CastEiJ and CastEiJ × C57BL/6 crosses was between 45 and 55% (*Table 1*).

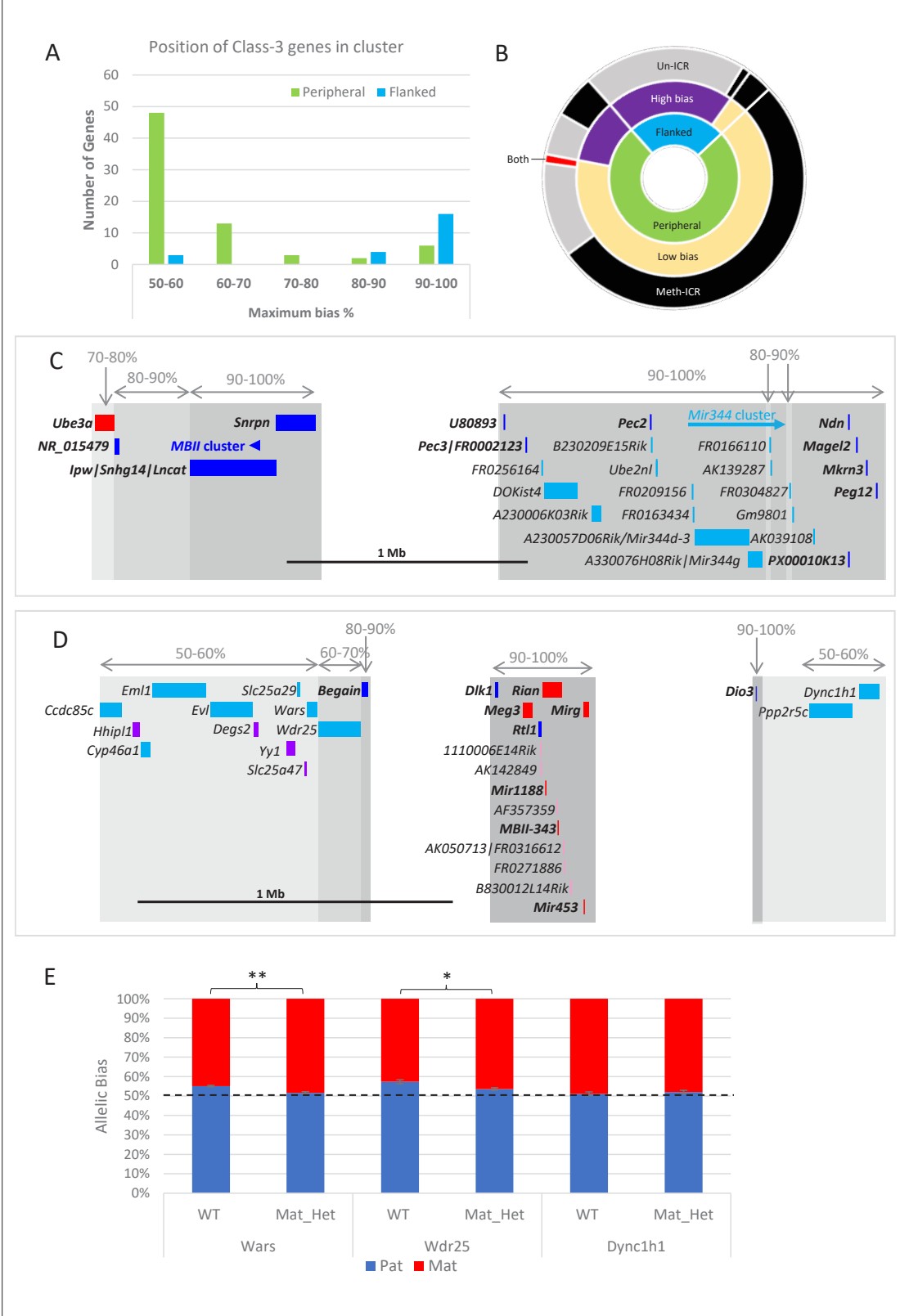

**Figure 4.** Weakly biased Class-3 genes are located at the periphery of known imprinted domains. (**A**) – Distribution of Class-3 biased genes in relation to known imprinted genes. Genes flanked by known imprinted genes are shown in turquoise and those peripheral to known imprinted genes are shown in green.(**B**) – Sunburst graph of a relationship between position in cluster, the extent of bias, and ICR methylation in Class-3 genes. Low bias = <70% expression from preferential chromosome, high bias = >70% expression from the preferential chromosome. Preferential expression from the methylated

*Figure 4 continued on next page*

*Figure 4 continued*

imprinting control region (ICR) chromosome is shown in black, preferential expression from the unmethylated ICR chromosome is shown in gray, and genes reported as being biased on both chromosomes depending on tissue or study are shown in red. (**C** and **D**) – Schematics of the *Snrpn* (**C**) and *Dlk1* (**D**) regions. Highly biased novel genes (80–100%) are located between known imprinted transcripts whereas low-biased genes (50–70%) are located at the periphery. Red boxes = known maternally expressed genes. Blue boxes = known paternally expressed genes. Pink boxes = novel maternally biased genes called by original studies. Turquoise boxes = novel paternally biased genes called by original studies. Blue arrow = cluster of imprinted MBII snoRNAs. Turquoise arrow *Mir344* cluster. (**E**) – Allele-specific expression analysis in peripheral genes in the *Dlk1* region in IG-DMR knockout mice. Female mice, heterozygous for IG-DMR knockout were crossed with male CastEiJ mice, and expression was assessed by pyrosequencing. Wildtype (n=5) and maternal heterozygote (n=6) expression biases were compared using an unpaired *t*-test ** p <0.01 * p < 0.5.

## Class-1

Of the eight novel singletons (Class-1) tested, four were biallelically expressed in all tissues and indeed three were expressed at very low levels. One gene, *Nhlrc1*, showed a consistent paternal bias across all postnatal brain tissues (*Table 1* and *Figure 5A*). It is noteworthy that this was the only Class-1 gene that was called in three of the original studies (*Figure 1C* and *Supplementary file 1c*). *Nhlrc1* lies 1.3 kb downstream from a known germline DMR and a Zfp57 binding site suggesting a plausible mechanism for biased gene expression at the region (*Proudhon et al., 2012*; *Strogantsev et al., 2015*). To see if this DMR persists postnatally, bisulfite pyrosequencing was performed on DNA from the P7 cerebellum and liver. In the cerebellum, the DMR is partially retained: the maternal allele is hypermethylated compared with the paternal allele in both BxC and CxB crosses (*Figure 5B*). However, in the liver where *Nhlrc1* shows no parental bias (*Figure 5—figure supplement 1A*), both alleles are hypermethylated (*Figure 5C*), indicating *Nhlrc1* expression is parentally biased only in tissues where the DMR is retained. This suggests a mechanism whereby a gDMR can influence tissue-specific imprinting postnatally.

## Class-2

Four of the 15 Class-2 genes tested fell below the expression threshold (*Table 1*). Of the other 11, six were biallelic in all tissues tested, two showed biased expression in the postnatal brain (*Pcdhb12* and *Wnk4*) and three had a weak maternal bias in the placenta (*Vat1*, *Rtn3,* and *Pla2g16*) (*Table 1*). *Pcdhb12* shows preferential expression from the maternal allele in all postnatal tissues which is consistent with the Datasets B and C (*Figure 5D*). This gene encodes protocadherin beta-12 and is in a cluster of protocadherin genes on chromosome 18 including *Pcdhb10* and *Pcdhb20* which were also called as biased in the original studies. Both of these genes were also tested: *Pcdhb10* was only expressed at very low levels and *Pcdhb20* showed biallelic expression (*Table 1*).

The other Class-2 gene which validated in the postnatal brain is *Wnk4*. This gene forms a novel cluster with *Vat1*, *Tmem106a*, and *Rdm1* on chromosome 11 that spans approximately 375 kb. Both validating tissues exhibit a weak bias. Indeed, *Wnk4* has a much stronger strain bias in both tissues in C57BL/6xCastEiJ hybrids which may be confounding the data (*Figure 5—figure supplement 2*). Further analysis of this gene in other reciprocal hybrid strains is necessary to confirm the nature of its bias. One of the other genes in the *Wnk4* novel cluster, *Vat1*, was one of three Class-2 genes that validated with maternal expression bias in the placenta (*Figure 5E*). The other two genes *Rtn3* and *Pla2g16* form a novel cluster with *Prdx5* (which is biallelic in all tissues tested) on Chromosome 19. All three genes with the maternal placental bias were originally called as being biased in neural tissue where no evidence for biased expression was found.

## Class-3

The 20 peripheral Class-3 genes assessed by allele-specific cDNA pyrosequencing had all been called as biased in brain tissues in the original studies and had maximum biases below 70%, except for *Ago2* which has a maximal maternal bias of 79.3%. In contrast with the putative-biased genes identified elsewhere in the genome, we found those peripheral to known imprinted clusters were more likely to validate. Nine of the 20 tested genes were validated in somatic tissues (*Adam23*, *Cox4i2*, *Bcl2l2*, *Tpx2*, *Smim17*, *Ifitm10*, *Wars*, *Wdr25*, and *Ago2* - *Table 1*). Of these *Bcl2l1* and *Ago2* showed a bias in all neural tissues tested (*Figure 6—figure supplements 1 and 2*). *Bcl2l1*, along with *Cox4i2* and *Tpx2* is located close to the known imprinted gene *Mcts2*. All three genes showed a paternal bias in at least five neural tissues. Interestingly, *Tpx2* was validated in all P60 tissues but not in the E16.5 or

**Table 1.** Table showing the summary of all the allele-specific pyrosequencing performed to validate putative-biased genes. Values show the mean expression (%) from the paternal allele of both reciprocal crosses to eliminate strain bias. Values above 55% are called as paternally biased (blue) and values below 45% are called as maternally biased (red). Assays with a strain bias of greater than 45:55 in more than one tissue are indicated in the 19th column. Genes that only validate in the placenta are called as Placental in the 20th column (Red = Maternal, Blue = Paternal).

| Genes | Chr. | Dataset | Direction previously reported | Class | e16.5 Plac. | e16.5 Liver | e16.5 Brain | P7 Cortex | P7 Hyp. | P7 Cb. | P7 Hipp. | P7 B.S | P60 Cortex | P60 Hyp. | P60 Cb. | P60 Hipp. | P60 B.S | Strain Bias | Validation status |
|---|---|---|---|---|---|---|---|---|---|---|---|---|---|---|---|---|---|---|---|
| *Class-1 (Novel Singletons)* | | | | | | | | | | | | | | | | | | | |
| L3mbtl1 | 2 | B | pat | 1 | - | - | - | 48.1 | 53.4 | 53.4 | 46.4 | 50.1 | 49.4 | 49.7 | 46.7 | 51.0 | 49.2 | | Biallelic |
| Ahi1 | 10 | B,D | pat | 1 | 52.8 | 46.5 | n/a | 47.0 | 51.6 | 50.9 | 52.8 | 46.9 | 48.6 | 49.4 | 50.0 | 52.4 | 51.3 | ✓ | Biallelic |
| Platr20 | 11 | A | pat | 1 | 51.7 | 49.9 | 50.0 | 49.9 | 50.4 | 50.1 | 50.4 | 50.2 | 49.8 | 49.8 | 50.1 | 50.2 | 49.7 | ✓ | Biallelic |
| Calm1 | 12 | B,D | pat | 1 | - | - | - | 54.7 | 49.5 | 51.4 | 45.9 | 43.4 | n/a | n/a | n/a | n/a | n/a | ✓ | Biallelic |
| Nhlrc1 | 13 | B, C, D | pat | 1 | - | - | - | 56.6 | 55.6 | 57.8 | 54.8 | 58.8 | 58.1 | 56.1 | 55.4 | 55.0 | 55.2 | ✓ | Paternal |
| Tnk1 | 11 | A | mat | 1 | - | - | - | - | - | - | - | - | - | - | - | - | - | | Low expression |
| Mlana | 18 | B | mat | 1 | - | - | - | - | - | - | - | - | - | - | - | - | - | | Low expression |
| Gm16299 | 19 | C | pat | 1 | - | - | - | - | - | - | - | - | - | - | - | - | - | | Low expression |
| *Class-2 (Novel Clusters)* | | | | | | | | | | | | | | | | | | | |
| Stx6 | 1 | C | mat | 2 | - | - | - | 50.7 | 49.2 | 50.3 | 49.8 | 50.1 | n/a | n/a | n/a | n/a | n/a | | Biallelic |
| Gabra5 | 7 | B,D | pat | 2 | - | - | - | 51.5 | 53.2 | 52.5 | 54.8 | 52.6 | 52.3 | 50.7 | 48.6 | 51.5 | 51.5 | | Biallelic |
| Wnk4 | 11 | C | mat | 2 | - | - | - | 50.1 | 52.0 | 45.3 | 50.6 | 44.0 | 48.1 | 44.6 | 50.9 | 45.4 | 48.1 | ✓ | Maternal |
| Vat1 | 11 | B | mat | 2 | 43.9 | 50.9 | 52.6 | 49.6 | 48.8 | 50.4 | 51.0 | 49.9 | 46.4 | 51.3 | 50.8 | 49.8 | 50.4 | ✓ | Placental |
| Rdm1 | 11 | A | mat | 2 | 49.4 | - | - | 48.8 | 48.8 | 50.1 | 50.2 | 50.7 | 46.9 | 49.5 | 50.5 | 50.1 | 48.7 | ✓ | Biallelic |
| Gaa | 11 | B,D | pat | 2 | 48.9 | 52.3 | 53.6 | 51.3 | 47.1 | 51.7 | 53.1 | 51.8 | 51.8 | 51.6 | 47.9 | 51.9 | 50.8 | ✓ | Biallelic |
| Pcdhb10 | 18 | D | mat | 2 | - | - | - | - | - | - | - | - | - | - | - | - | - | | Low expression |
| Pcdhb12 | 18 | B,C | mat | 2 | - | - | - | 40.5 | 41.2 | 41.0 | 39.4 | 41.8 | 42.6 | 42.7 | 42.1 | 43.2 | 44.9 | ✓ | Maternal |
| Pcdhb20 | 18 | B,C | pat | 2 | - | - | 52.7 | 51.5 | 51.6 | 52.0 | 52.6 | 53.1 | 51.5 | 52.3 | 51.6 | 51.6 | 50.1 | ✓ | Biallelic |
| Prdx5 | 19 | B | pat | 2 | 46.9 | 51.1 | 49.4 | 49.0 | 50.4 | 48.8 | 49.4 | 49.9 | 49.2 | 50.2 | 50.0 | 48.3 | 49.2 | ✓ | Biallelic |
| Rtn3 | 19 | D | pat | 2 | 44.6 | 49.9 | 50.7 | 49.4 | 47.3 | 49.5 | 49.5 | 49.2 | 49.1 | 50.9 | 49.7 | 50.4 | 50.4 | | Placental |
| Pla2g16 | 19 | B | mat | 2 | 40.3 | 51.6 | 50.6 | 51.7 | 49.7 | 48.9 | 48.5 | 50.7 | 51.5 | 51.2 | 49.2 | 46.8 | 48.3 | ✓ | Placental |
| Mr1 | 1 | C | mat | 2 | - | - | - | - | - | - | - | - | - | - | - | - | - | | Low expression |
| BC034090 | 1 | C | mat | 2 | - | - | - | - | - | - | - | - | - | - | - | - | - | | Low expression |

*Table 1 continued on next page*

*Table 1 continued*

| Genes | Chr. | Dataset | Direction previously reported | Class | e16.5 Plac. | e16.5 Liver | e16.5 Brain | P7 Cortex | P7 Hyp. | P7 Cb. | P7 Hipp. | P7 B.S | P60 Cortex | P60 Hyp. | P60 Cb. | P60 Hipp. | P60 B.S | Strain Bias | Validation status |
|---|---|---|---|---|---|---|---|---|---|---|---|---|---|---|---|---|---|---|---|
| Tmem106a | 11 | A | mat | 2 | - | - | - | - | - | - | - | - | - | - | - | - | - | | Low expression |
| *Class-3 (Close to known imprinted genes)* | | | | | | | | | | | | | | | | | | | |
| Adam23 | 1 | A,B,C,D | pat | 3 | 48.0 | 52.0 | 59.1 | 56.5 | 57.8 | 56.5 | 58.0 | 53.7 | 55.6 | 58.7 | 53.5 | 56.5 | 54.1 | | Paternal |
| Mcts2 | 2 | A,B,C | pat | K | 77.2 | 82.9 | 64.7 | 71.8 | 80.2 | 73.1 | 77.3 | 80.4 | 85.7 | 78.3 | 70.6 | 87.8 | 84.2 | | Paternal |
| Cox4i2 | 2 | C | pat | 3 | 49.3 | - | - | 69.8 | 51.0 | 56.6 | 60.0 | 56.5 | 55.3 | 55.0 | 52.8 | 56.8 | 54.9 | | Paternal |
| Bcl2l1 | 2 | A,B,C,D | pat | 3 | 49.5 | 50.2 | 61.6 | 60.9 | 61.8 | 58.3 | 59.5 | 58.8 | 59.9 | 61.4 | 57.4 | 59.2 | 59.0 | | Paternal |
| Tpx2 | 2 | C | pat | 3 | - | 49.7 | 49.1 | 53.4 | 52.6 | 50.7 | 53.6 | 51.4 | 56.4 | 55.4 | 64.6 | 62.5 | 61.1 | ✓ | Paternal |
| Herc3 | 6 | A,B,C,D | mat | K | 47.0 | 45.8 | 43.1 | 45.7 | 40.7 | 40.5 | 49.9 | 32.4 | 44.0 | 29.5 | 39.0 | 42.3 | 24.4 | ✓ | Maternal |
| Fam13a | 6 | B,D | mat | 3 | - | 46.1 | 53.2 | 47.9 | 47.1 | 51.3 | 49.4 | 48.4 | n/a | n/a | n/a | n/a | n/a | ✓ | Biallelic |
| Zfp78 | 7 | B | both | 3 | - | - | - | - | - | - | - | - | - | - | - | - | - | | Low expression |
| Smim17 | 7 | B,D | mat | 3 | - | - | 38.3 | 48.8 | 35.5 | 50.9 | 44.0 | 44.5 | 45.2 | 43.2 | 52.1 | 50.8 | 49.3 | | Maternal |
| Peg3 | 7 | A,B,C,D | pat | K | 96.3 | 99.2 | 99.6 | 92.4 | 92.8 | 94.1 | 94.3 | 95.1 | 97.5 | 98.0 | 98.0 | 97.3 | 97.7 | ✓ | Paternal |
| Zfp954 | 7 | B | mat | 3 | - | - | - | - | - | - | - | - | - | - | - | - | - | | Low expression |
| Zfp773 | 7 | B | mat | 3 | - | - | - | - | - | - | - | - | - | - | - | - | - | | Low expression |
| Zfp772 | 7 | B | mat | 3 | - | - | - | - | - | - | - | - | - | - | - | - | - | | Low expression |
| Clcn4-2 | 7 | B | mat | 3 | 44.6 | 54.5 | 48.5 | 48.9 | 48.8 | 49.9 | 49.1 | 48.0 | 49.5 | 50.7 | 51.0 | 49.6 | 50.7 | ✓ | Placental |
| Ifitm10 | 7 | C,D | mat | 3 | 46.9 | 52.3 | 48.7 | 47.5 | 41.5 | 53.0 | 49.4 | 45.0 | 49.0 | 43.2 | 43.6 | 50.7 | 47.0 | ✓ | Maternal |
| Ctsd | 7 | B,D | mat | 3 | 46.8 | 47.8 | 50.1 | 50.0 | 49.0 | 48.4 | 48.9 | 49.8 | 49.3 | 50.2 | 49.0 | 50.5 | 49.5 | | Biallelic |
| Evl | 12 | B | pat | 3 | 46.2 | 48.3 | 50.2 | 48.7 | 51.3 | 51.1 | 50.2 | 49.5 | 50.9 | 50.9 | 52.6 | 50.8 | 50.6 | ✓ | Biallelic |
| Slc25a29 | 12 | B | pat | 3 | 18.9 | 45.5 | 52.3 | 53.9 | 56.2 | 54.5 | 53.3 | 54.2 | 52.8 | 53.8 | 50.6 | 53.6 | 53.2 | ✓ | Placental |
| Wars | 12 | C | pat | 3 | 45.2 | 52.5 | - | 53.9 | 56.2 | 54.5 | 53.3 | 54.2 | 52.8 | 53.9 | 50.6 | 53.6 | 53.2 | | Paternal |
| Wdr25 | 12 | B,D | pat | 3 | - | - | - | 50.7 | 52.9 | 52.0 | 50.6 | 48.7 | 51.8 | 59.8 | 49.9 | 51.7 | - | ✓ | Paternal |
| Dlk1 | 12 | A,B,C,D | pat | K | 95.7 | 87.9 | 93.7 | 92.5 | 92.7 | 94.6 | 90.2 | 95.0 | 89.5 | 95.6 | 93.2 | 86.9 | 95.2 | | Paternal |
| Ppp2r5c | 12 | B,C | pat | 3 | 52.4 | 51.3 | 48.5 | 48.6 | 49.3 | 54.0 | 50.6 | 48.2 | 50.6 | 48.7 | 45.2 | 51.0 | 49.3 | ✓ | Biallelic |
| Dync1h1 | 12 | B,C | pat | 3 | 46.9 | 49.8 | 50.4 | 50.7 | 50.7 | 51.1 | 49.9 | 50.4 | 51.0 | 50.4 | 50.1 | 50.1 | 49.8 | | Biallelic |
| Ago2 | 15 | A,B,C,D | mat | 3 | 47.7 | 50.7 | 30.1 | 24.1 | 25.8 | 28.8 | 28.7 | 19.3 | 25.9 | 28.7 | 38.0 | 33.3 | 24.6 | | Maternal |

*Table 1 continued on next page*

*Table 1 continued*

| Genes | Chr. | Dataset | Direction previously reported | Class | e16.5 Plac. | e16.5 Liver | e16.5 Brain | P7 Cortex | P7 Hyp. | P7 Cb. | P7 Hipp. | P7 B.S | P60 Cortex | P60 Hyp. | P60 Cb. | P60 Hipp. | P60 B.S | Strain Bias | Validation status |
|---|---|---|---|---|---|---|---|---|---|---|---|---|---|---|---|---|---|---|---|
| *Ampd3* | 7 | B | mat | K | 23.4 | 47.5 | 51.4 | 48.6 | 48.5 | 48.3 | 51.8 | 50.9 | 50.1 | 46.7 | 50.9 | 49.4 | 51.5 | ✓ | Placental |
| *Gab1* | 8 | A | pat | K | 74.9 | 50.4 | 49.8 | 52.7 | 49.6 | 48.6 | 49.4 | 51.0 | 47.1 | 49.6 | 48.8 | 50.1 | 50.2 | | Placental |

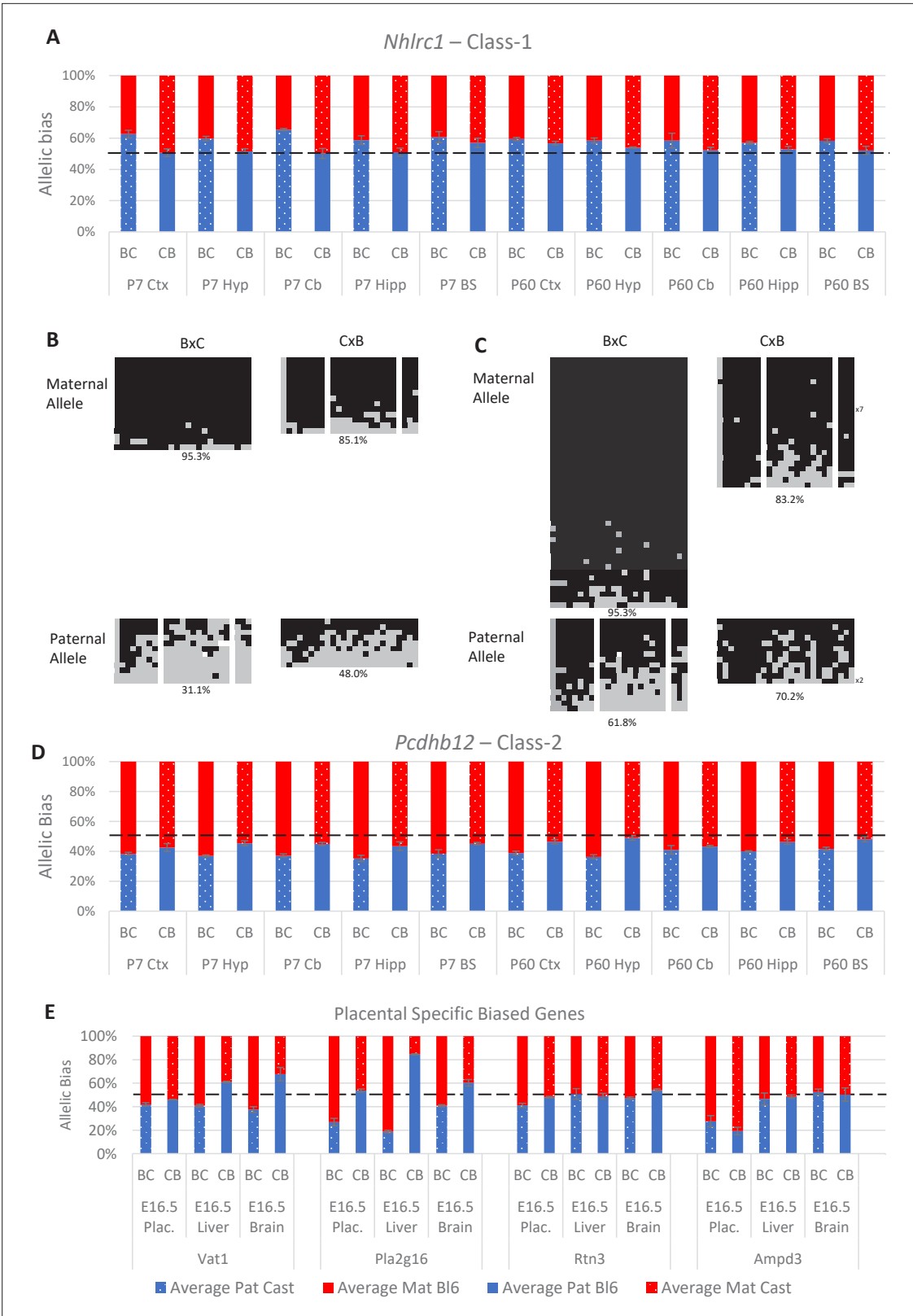

**Figure 5.** Experimentally validated Class-1 and Class-2 genes. (**A**) – *Nhlrc1* (Class-1) is paternally biased in all postnatal neuronal tissues tested. (**B-C**) Bisulfite sequencing analysis in P7 tissues.(**B**) – Cerebellum, (**C**)- Liver. Each line represents a different clone of bisulfite sequencing derived from two BxC animals and two CxB animals. Numbers of identical clones sequenced are indicated to the right. Black = methylated CpG and Gray = unmethylated CpG, white = CpG absent from clone. Percentage of methylated CpGs from all clones is indicated underneath.(**D**) – *Pcdhb12* (Class-2)

*Figure 5 continued on next page*

*Figure 5 continued*

is maternally biased in all postnatal tissues tested. (**E**) – Three Class-2 genes show a maternal bias in e16.5 placenta: *Vat1, Pla2g16,* and *Rtn3*. These biases are weaker than seen in *Ampd3* which is imprinted in the placenta (*Schulz et al., 2006*). Allele-specific expression graphs (**A, D and E**) show mean expression (%) from the paternal allele (deep blue) and maternal allele (red) in C57BL/6 × CastEiJ (BC) and four CastEiJ × C57BL/6 (CB) crosses. Castaneus allele is denoted by a spotted pattern. Standard error of the mean is shown n=3 or 4. Data are normalized to gDNA.

The online version of this article includes the following figure supplement(s) for figure 5:

**Figure supplement 1.** Coro1c pseudogene is almost exclusively expressed from the castaneus allele.

**Figure supplement 2.** Allelic bias in *Wnk4* novel cluster.

**Figure supplement 3.** Expression levels of genes tested by allele-specific pyrosequencing.

P7 material (*Table 1* and *Figure 6—figure supplement 1*) suggesting the bias strengthens over time postnatally.

Of the six genes tested that are located close to the *Peg3* cluster, four could not be tested due to low expression. In contrast, *Smim17* (*Gm16532*) is preferentially expressed from the maternal allele in five neural tissues (*Figure 6—figure supplement 3*). *Smim17* bias is strongest in the P7 hypothalamus (64.7%) but is reduced to 56.7% by P60. A maternal bias is also detected in the P7 hippocampus and brain stem but is lost by P60, together indicating the *Smim17* bias reduces over time (*Table 1*).

Six genes were tested that are located at the periphery of the *Dlk1/Dio3* cluster on Chromosome 12. *Wdr25* was biallelically expressed in all tissues except the P60 hypothalamus where 59.7% of expression is from the paternal allele (*Figure 6D*). Unlike *Wdr25*, *Wars* expression was consistently higher from the paternally inherited allele in all somatic tissues however, the bias was only above the 55% cut-off in two tissues: e16.5 brain (55.1%) and P7 hypothalamus (56.2%) (*Figure 6C*). The most peripheral genes tested on both the proximal (*Evl*) and distal (*Ppp2r5c* and *Dync1h1*) side of the cluster were called as paternally biased in the original studies but biallelic in all tissues we assessed (*Figure 6A and F* and *Table 1*). *Slc25a29* was called as paternally biased in ARN and DRN in Dataset B, but we found it to be biallelic in all neural tissues. Taken together our data suggest the weak biases observed at the periphery of known imprinted domains are tissue- and stage-specific.

Interestingly, *Slc25a29* showed a very strong maternal bias in the placenta (81.1% - *Figure 6B*). To determine if this biased expression is regulated by the *Dlk1/Dio3* imprinting control region, we again made use of the IG-DMR knockout mouse model (*Lin et al., 2003*), Male and female mice heterozygous for the IG-DMR deletion were crossed with CastEiJ mice and placentas were collected at E15.5. Allele-specific pyrosequencing revealed maternal and paternal heterozygotes to both the same degree of maternal bias as wildtype litter mates indicating that *Slc25a29* imprinting in the placenta is not under the control of the IG-DMR (*Figure 6G*).

## Discussion

In order to discuss weak parent-of-origin expression bias, it is first necessary to define canonical imprinting. Historically, imprinted expression was defined as monoallelic but as the sensitivity of methods quantifying ASE has improved it has become apparent that many imprinted genes show low-level expression from the repressed allele. Within a single canonically imprinted cluster, the extent of bias can vary greatly between genes. For example, in the *Dlk1/Dio3* region >98% of *Meg3/Gtl2* expression arises from the maternal allele, and 84–99% of *Dlk1* expression arises from the paternal chromosome, but for *Dio3*, 80% of expression is from the paternal allele in the embryo (*Tsai et al., 2002*). However, *Dio3* is still considered to be an imprinted gene as this bias is established by the ICR for this region and when the ICR is deleted from the maternal chromosome imprinting is lost from the entire region and *Dio3* shows 50:50 expression from both alleles (*Lin et al., 2003*).

We, therefore, propose to define 'canonical imprinting' as genes that are expressed predominantly (>70%) from one allele in a parent-of-origin specific manner in at least one tissue. We have taken a 70% cut-off as this was used previously by others (*Andergassen et al., 2017*). Canonical imprinted genes are also under the control of a differentially methylated ICR that is established during gametogenesis with expression returning to biallelic levels or completely lost on perturbation of the ICR. Genes with a weak bias at the periphery of known imprinted domains that lose the expression bias on the loss of the ICR we term 'weak canonical imprinting'. The term 'non-canonical imprinting' is often

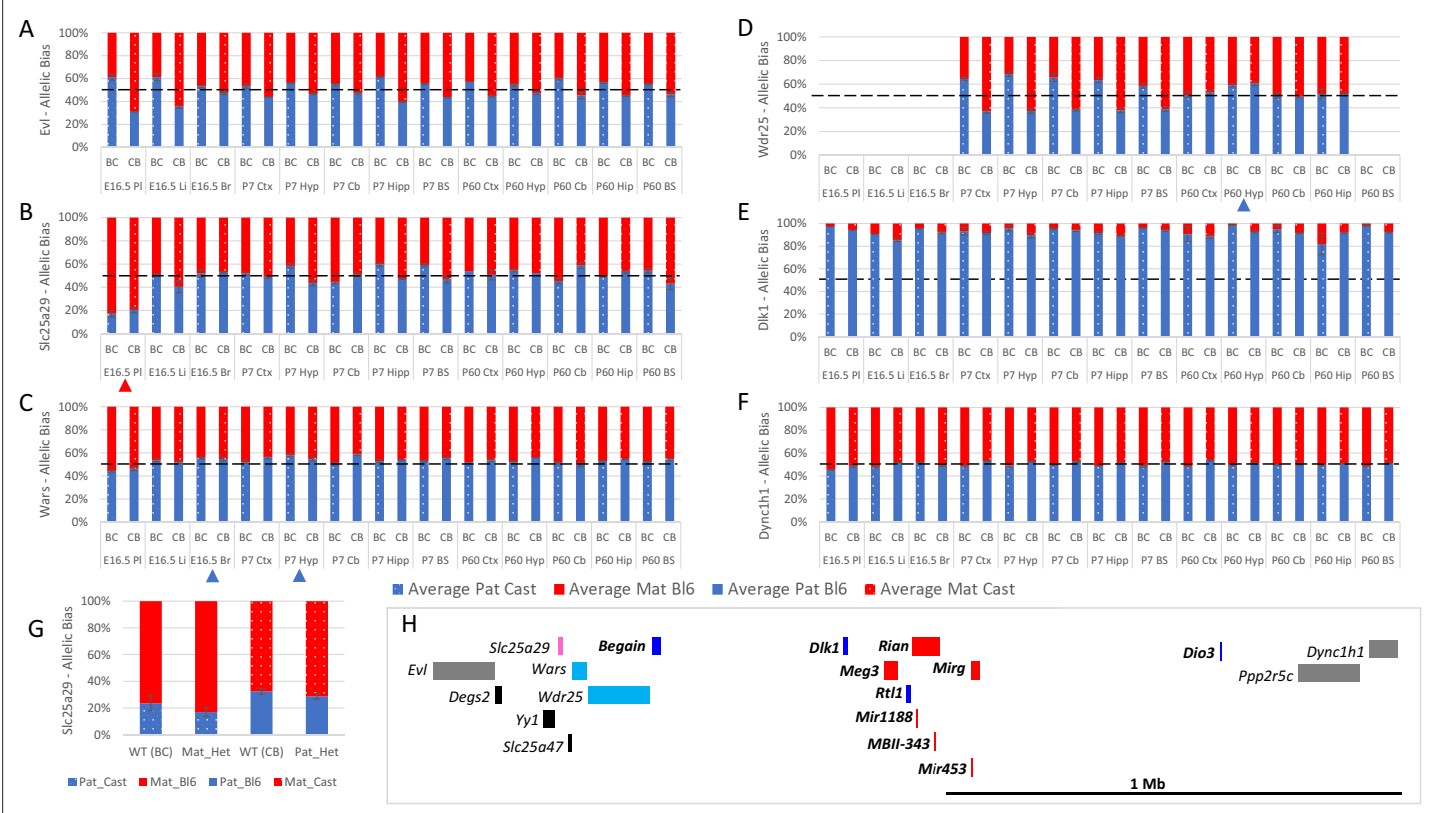

**Figure 6.** Allele-specific expression analysis of the *Dlk1* domain. (*A–F*) *Allelic bias in Evl* (**A**), *Slc25a29* (**B**), *Wars* (**C**) *Wdr24* (**D**) *Dlk1* (**E**), and *Dync1h1* (**F**) Graphs show mean expression (%) from the paternal allele (blue) and maternal allele (red) C57BL/6 × CastEiJ (BC) and four CastEiJ × C57BL/6 (CB) crosses. Castaneus allele is denoted by a spotted pattern. Standard error of the mean is shown, n=3 or 4. Tissues with a mean bias greater than 45:55 are indicated by arrow heads. Amplification bias was assessed in genomic DNA and the data are corrected. (**G**) – Imprinting of Slc25a29 in e15.5 placenta is not under the control of the IG-DMR. WT (BC) = maternal allele is wildtype for the IG-DMR paternal allele is CastEiJ (n=5). Mat_Het = maternal allele has IG-DMR deletion and the paternal allele is CastEiJ (n=6). WT (CB)=paternal allele is wildtype for the IG-DMR maternal allele is CastEiJ (n=5). Pat_Het = Paternal allele has IG-DMR deletion and the maternal allele is CastEiJ (n=7). (**H**) Schematic of the validated expression data in the *Dlk1* region. Red boxes = known maternally expressed genes. Blue boxes = known paternally expressed genes. Pink boxes = novel validated maternally biased genes. Turquoise boxes = novel validated paternally biased genes. Gray boxes = biallelically expressed genes.

The online version of this article includes the following figure supplement(s) for figure 6:

**Figure supplement 1.** Allelic bias in the *Mcts2* region.

**Figure supplement 2.** Allelic bias (%) in *Adam23* (**A**), *Ifitm10* (**B**), and *Ago2* (**C**).

**Figure supplement 3.** Allelic bias in *Peg3* region.

used to refer to those genes for which imprinting is established in the germline via regions of differential of H3K27me3 (DMKs) leading to tissue-specific imprinting in the extra-embryonic compartment, such as *Sfmbt2, Smoc1, and Gab1* (*Inoue et al., 2017*; *Okae et al., 2012*; *Hanna et al., 2019*). Finally, genes with a weak bias of less than 70% in tissues and with no known mechanism for the establishment of the bias we call 'parental-origin-specific biased genes'.

Although many parental-origin-specific biased genes were identified in the brain in the original studies, only two of the Class-1 or 2 genes tested here were validated in more than two tissues (*Nhlrc1* and *Pcdhb12*). Our analysis also revealed a little overlap of novel weakly biased genes between the four datasets detailed above, even when compared between the same tissues. This lack of reproducibility could be due to several factors. Firstly, expression levels can influence ASE calling: the lack of read depth in lowly expressed genes may erroneously lead to genes being called as biased, because a small difference in read numbers produces a larger bias in weakly expressed transcripts. Of the 23 Class-1 or 2 genes tested by pyrosequencing, seven could not be confirmed due to low expression levels. However, re-analysis of the RNA-seq data from three of the original studies shows weakly expressed genes are not overrepresented in the novel biased genes (*Figure 2—figure supplement*

*1*). Furthermore, when the expression levels are extracted for each of the genes tested by pyrose-quencing, we see no correlation between expression levels and validation (***Figure 5—figure supplement 3***). Expression levels are consistent between studies, especially in the same tissue, indicating the lack of reproducibility between studies is not due to differing expression. Second, the number of biological replicates can greatly influence the number of biased genes called. We found that analysis of datasets with only one or two replicates (Datasets A and E) leads to a higher number of false positive novel biased genes being called compared with datasets with >6 replicates (Datasets B and C). Third, the influence of genetic background on gene expression needs to be considered. We found a greater overlap between the original studies with genes called as strain biased by ISoLDE than those called with a parental bias. Moreover, 22 of the genes we experimentally tested had a strain bias in two or more tissues compared with 16 genes with parental origin bias in one or more tissues. Finally, the method for calling ASE greatly affects the results: minimal overlap was observed between the same data analyzed by two different methods (***Figure 2A–C***). It is noteworthy that the two Class-1 or 2 genes that experimentally validated in most tissues were both called by two or more of the original studies suggesting a more accurate picture of ASE may be achieved by combining two calling methods. Taken together, these observations highlight the need for careful planning before embarking on projects to call parental-bias, stringent filtering for low expression and strain effects, and validation via an independent method to confirm findings.

Most of the unique novel genes identified in the original studies are in regions where imprinting has not previously been identified. Although only 6 of the 23 we tested validated experimentally (three in the placenta) there may be others that display a true parent-of-origin bias. None of the Class-1 and 2 genes that were validated show a bias greater than 60:40. Such weak biases can be explained in different ways (***Figure 7***). First, a weak bias is seen in each cell in the population (***Figure 7A***). Second, bias may be due to true imprinting in a subset of cells within the population which could be random, clonal, or cell type-specific (***Figure 7B and C***). If this is due to canonical imprinting, we would expect to see a DMR. Indeed, at the *Nhlrc1* locus, we found that a previously reported Zfp57-bound, germline DMR (***Proudhon et al., 2012***; ***Strogantsev et al., 2015***) is partially retained in postnatal neural tissues indicating that its expression bias could be regulated by canonical imprinting mechanisms. We assessed the genomic intervals with other validated novel genes for germline DMRs (gDMRs) or Zfp57 binding using previously published data (***Strogantsev et al., 2015***; ***Kobayashi et al., 2012***). No gDMRs or Zfp57 binding sites were reported in the *Pcdhb* cluster. In the novel cluster between *Prdx5* and *Pla2g16* there are three oocyte-methylated gDMRs (***Kobayashi et al., 2012***). However, none of them overlap with the biased genes or a Zfp57-bound region (***Strogantsev et al., 2015***). Interestingly, there are two oocyte-methylated gDMRs in the interval between *Wnk4* and *Rdm1*, both of which overlap with *Wnk4*: one over the promoter and one over exons 5 and 6 (***Figure 5—figure supplement 2***; ***Kobayashi et al., 2012***). As *Wnk4* is maternally biased it is hard to reconcile with methylation at the maternal promoter since most maternally methylated promoters lead to repression of the maternal allele. However, the second DMR overlaps with a lncRNA (*Gm11615*): further investigation is needed to establish if this transcript is reciprocally biased and offers a possible mechanism for the observed *Wnk4* bias. Finally, bias could be due to a skew towards activating one allele in random monoallelic expression (***Figure 7D***). The 22 *Pcdhb* isoforms exhibit allelic exclusion and are mono-allelically expressed in a stochastic and combinatorial fashion in individual neurons so each cell expresses two genes from the maternally inherited chromosome and two from the paternal copy (***Hirayama and Yagi, 2017***). However, *Pcdhb12* showed a maternal bias in all postnatal tissues tested. This may reflect interindividual random bias: but, of the 60 tissue samples tested, 54 showed a maternal bias in *Pcdhb12* and only six showed a paternal bias, indicating there are more neurons expressing the maternally inherited copy of the gene than randomly expected. This suggests that the *Pcdhb12* bias does not reflect a maternal bias in each cell but rather a population bias within the brain of individuals: single cell analysis of neurons is needed to confirm this hypothesis.

Most novel genes called in more than one study reside in or close to known imprinted regions. These genes were also more likely to validate (11 out of 20) but again, it is not known on a population level if these biases occur in every cell or are the result of complete imprinting is a subset of cells. We find that parental bias is stronger for known and novel genes within imprinted clusters than those at the periphery of the cluster. At the periphery, biased genes show temporal and tissue-specific changes: *Tpx2* only validates in P60 samples whereas *Smim17*'s bias decreases at this later stage

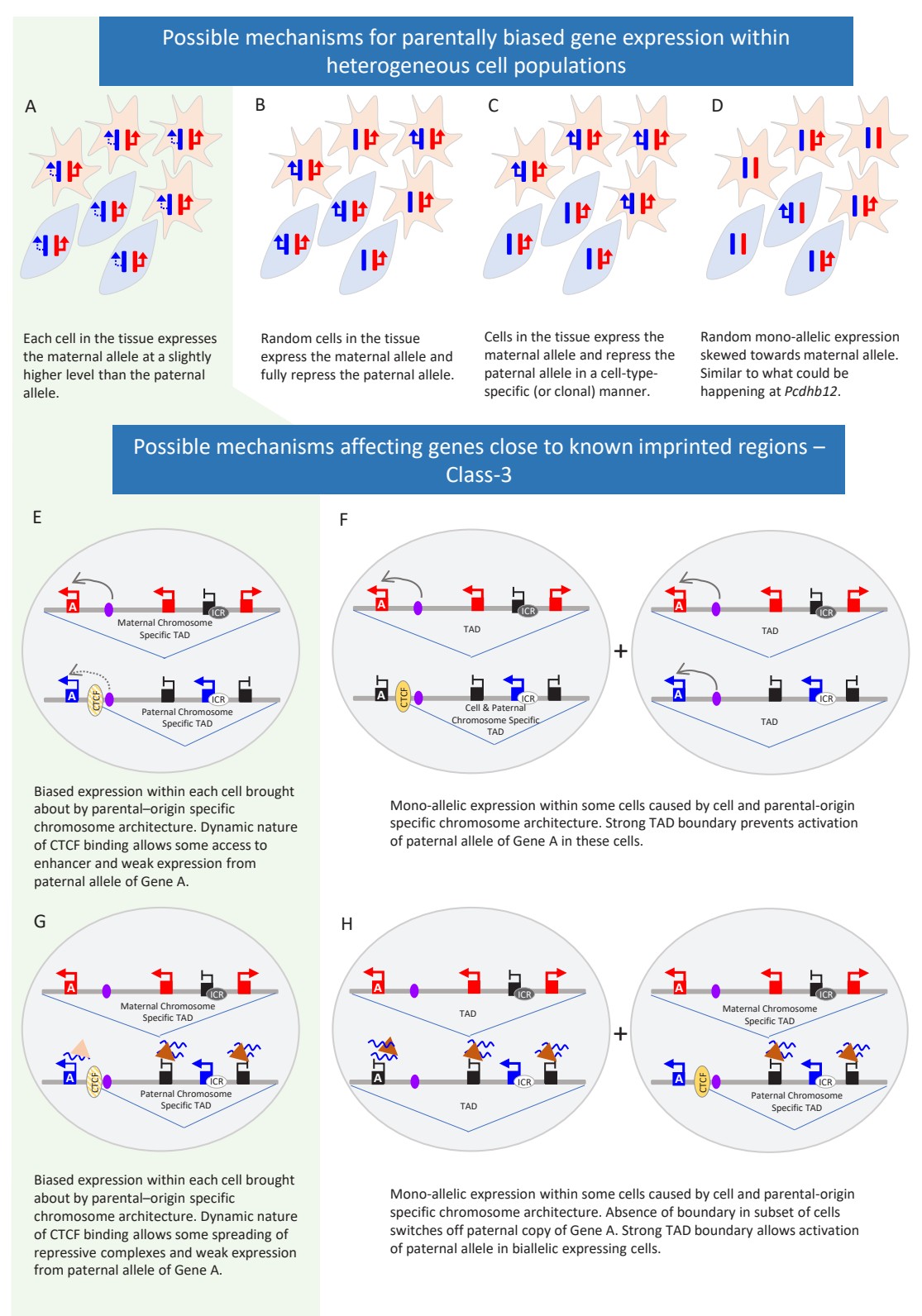

**Figure 7.** Possible mechanisms behind parent-of-origin expression biases in tissues. (**A-D**) Scenarios causing biased expression in heterogenous cell populations: bias occurs in every cell in the tissue (**A**), random imprinting in the subset of cells (**B**), cell-type specific imprinting (**C**) or random monoallelic expression that is skewed towards one allele (**D**). (**E-H**) Possible Mechanisms behind parent-of-origin biases at the periphery of imprinted domains.

showing biases are dynamic. We have found that weak Class-3 genes are more likely to be repressed on the chromosome with the unmethylated ICR where repression of known imprinted genes relies on repressive histone marks. Perez et al. previously showed that biases decrease towards the edge of imprinted clusters. Together, these observations suggest the influence of ICRs diminishes with distance and that differences in the local environment on each chromosome may play a role in biased gene expression. One possibility is that differential 3D architecture between the two chromosomes is contributing to the bias. It has long been known that the *Igf2/H19* cluster shows different conformation on each parental chromosome (*Murrell et al., 2004*) and a more recent single-cell 3D analysis found eight known imprinted regions have parental-origin-specific TADs in the postnatal cortex and hippocampus (*Tan et al., 2021*). TADs are flanked by convergent CTCF sites and were originally believed to be stable domains that demarcate regions of shared regulation (*Dixon et al., 2012*; *Rao et al., 2014*). However, recent findings show CTCF binding is dynamic (*Hansen et al., 2017*) and the boundaries of TADs can vary between cells (*Nagano et al., 2013*; *Szabo et al., 2020*). It is also known that the expression of an imprinted ncRNA from one chromosome can recruit a polycomb repressive complex to repress other genes *in cis* (*Terranova et al., 2008*). It is, therefore, possible that in imprinted regions, parental chromosome-specific conformations are affecting the expression of peripheral genes either by preventing the spread of repressive marks to the periphery or by reducing the enhancer/silencer interactions on the weaker allele (*Figure 7E–H*). Such mechanisms could bring about weak bias in all cells or complete imprinting in a subset and are likely locus dependent; higher resolution analysis at the end of clusters and in single cells is needed to address this.

We have shown that the number of novel imprinted genes in the genome has been over-estimated by RNA-seq. However, some weakly biased genes do exist. We confirmed the weakly biased imprinting of the novel singleton *Nhlrc1* and many Class-3 genes. What is the functional relevance of such weak biases? As strain bias has a greater effect on expression than parent-of-origin bias then imprinting per se, may not be functionally relevant, especially if the bias is occurring within each cell. If on the other hand, the bias is due to canonical imprinting in a subset of cells, then it may be important. One known weak canonically imprinted gene is *Th* (*Schulz et al., 2006*), which was picked up by two of the studies (*Bonthuis et al., 2015*; *Crowley et al., 2015*) and shows 50–60% maternal expression bias. Importantly, this gene was found to be monoallelic in a subset of cells and be associated with a behavioral phenotype depending on maternal or paternal inheritance (*Bonthuis et al., 2015*). It is, therefore, possible that other weakly biased genes are robustly imprinted in certain cell types and where their monoallelic expression is functionally important. Indeed, the *Bcl2l1* is a weak paternally biased gene in the *H13/Mcts2* imprinted domain, deletion of which leads to loss of certain neuron types and reduction in brain mass upon paternal but not maternal transmission (*Perez et al., 2015*). In conclusion, genomic imprinting is not as widespread in the genome as recent studies have claimed. However, at the periphery of known imprinted clusters, weak parent-of-origin specific bias appears to be conferred on some genes. Whether all these weakly bias genes are functionally relevant or if some are simply innocent bystanders of local allele- and tissue-specific environments remains to be established.

## Methods
### Comparison of methods used in the four studies chosen for analysis

For the best comparison, we ignored all non-mouse data, as well as imprinting data from the mouse X chromosome, which was only analyzed in one of the studies (*Bonthuis et al., 2015*) as well as any non-imprinting work performed. Crowley et al., used CastEiJ, PWK/PhJ, and WSB/EiJ mice in reciprocal crosses of all combinations whereas the other three studies all used C57BL/6 x CastEiJ reciprocal mouse crosses (*Supplementary file 1a*; *Babak et al., 2015*; *Bonthuis et al., 2015*; *Crowley et al., 2015*; *Perez et al., 2015*). As RNA-seq starting material, Perez and colleagues used mouse cerebellum at postnatal day (P)8 and P60 (Dataset C) (*Perez et al., 2015*). Bonthuis et al., used mouse liver and skeletal muscle, as well as two highly specific brain regions, the arcuate nucleus (ARN) of the hypothalamus and the dorsal raphe nucleus (DRN) of the midbrain, at 8–10 weeks of age (Dataset B) (*Bonthuis et al., 2015*). Crowley and colleagues used the whole brain at P23 for their imprinting analysis (Dataset D) (*Crowley et al., 2015*). The Babak et al., study generated an imprinting atlas from 23 tissues from between 35- and 45 days' post-partum (including whole brain and six brain regions) and

three fetal tissues, in addition to reanalyzing data from five previous studies (Dataset A) (*Babak et al., 2015*). Details of the four different studies can be found in *Supplementary file 1a*.

Although all studies used the same methods for RNA-seq (Illumina Tru-Seq RNA Kit v2), each group used a different number of replicates. The Crowley and Perez studies used six males and six females for each reciprocal cross in each tissue. Whereas Bonthuis and colleagues used eight or nine replicate females for each cross and tissue. Babak et al., pooled tissues dissected from two males for some tissues so only one library was generated for each reciprocal cross in most cases. All four studies used the Illumina HiSeq 2000 sequencing platform. The Bonthuis and Perez studies generated 59 bp single-end data and Babak et al., and Crowley et al., generated 90 bp and 100 bp paired-end data, respectively.

Three of the studies (Babak, Crowley, and Perez) used previously annotated SNPs whilst Bonthuis et al., performed RNA-seq on C57BL/6 and Castaneus parents, allowing them to identify 400,820 SNPs high confidence SNPs between the two strains. All four studies also used different statistical analyses to call allele-specific expression. Babak et al. quantified ASE for each gene by counting uniquely mapped read pairs within the gene (including introns) that overlapped at least one SNP and allowed the parent of origin to be determined. They then estimated ASE from the cumulative binomial distribution with the random expectation set to 50% (equal expression from both alleles). The log10 of the least significant p-value between the two reciprocally crossed tissue is taken as the imprinting score with the paternal bias set as negative and maternal expression as positive. The Bonthuis study tallied the C57BL/6 and Castaneus reads for all SNPs across each informative gene adjusting for reads that contained more than one SNP. They then used generalized linear modeling with false discovery rate (FDR) estimation centered upon a permutation-based approach. To reduce the variance in the data related to sample and strain effects they contrasted the full model: (allele counts ~sample + strain +parent) with the reduced model (allele counts ~sample + strain). For each dataset, they used a p-value that yielded an FDR of 1%. Crowley et al., called ASE when a paired-end read overlapped at least one SNP or indel that was heterozygous between the paternal and maternal strains. They then used the method detailed in *Zou et al., 2014* to jointly model the total number of reads (TReC) and the number of allele-specific reads (ASE) of each gene, which, they claim, significantly boosts power for detecting strain and particularly parent-of-origin effects. Finally, the Perez study used an in-house pipeline, the Bayesian regression allelic imbalance model (BRAIM), which accounts for all sources of variability in the experimental design i.e., cross, sex, and age.

A critical analysis of using RNA-seq to infer imprinted expression highlighted the need for validation via an independent method (*DeVeale et al., 2012*). However, only three of the studies included such validation. One of the studies (*Perez et al., 2015*) conducted a full quantitative validation by pyrosequencing of all novel genes. They found nine false positives, which are not included in our meta-analysis. Bonthuis and colleagues validated only a subset of 18 novel genes by pyrosequencing, of which 8 failed (*Bonthuis et al., 2015*). Since this approach was not very systematic, testing only 12% of all novel genes, we included the original RNA-seq results including the false positives in our meta-analysis. Babak and colleagues validated most of the novel genes by pyrosequencing but in only one of all respective positive tissues (*Babak et al., 2015*). They only made RNA-seq data available for novel genes that either passed validation (13 genes) or could not be tested (12 genes), so we could not include other results. The final numbers of ASE autosomal genes reported for each study were as follows: Babak et al., - 125 (Dataset A), Bonthuis et al., - 210 (Dataset B), Perez et al., - 115 (Dataset C), and Crowley et al., – 95 (Dataset D) (*Babak et al., 2015*; *Bonthuis et al., 2015*; *Crowley et al., 2015*; *Perez et al., 2015*).

## Meta-analysis of previous data

Co-ordinates from the Datasets A and B were converted to mm10 using the LiftOver function at UCSC (Kent et al.). Any overlapping genes with different names were individually assessed and merged if found to represent the same transcript. Overlaps between studies were then assessed.

The maximum bias from each gene was assigned into 1 of 5 bins (50-60, 60-70, 70-80, 80-90, 90-100). For Dataset C these were taken straight from elife-07860-supp1-v2.xlsx (Table G). For the Dataset A, read counts for the reciprocal crosses were taken from the original source data. The bias for each gene in every tissue was calculated by:

$$Paternal\ reads\ (BxC + CxB)\ /Total\ reads\ (BxC + CxB)$$

or

$$Maternal\ reads\ (BxC + CxB)\ /Total\ reads\ (BxC + CxB)$$

For Dataset B, the maternal bias for each gene in every tissue was calculated as:

$$\left(Fold\ Change\ \left(\tfrac{Maternal}{Paternal}\right)\ /1 +\ Fold\ Change\ \left(\tfrac{Maternal}{Paternal}\right)\right)\ x\ 100$$

For Dataset D, the maximum bias was calculated by taking the mean paternal and maternal bias from each reciprocal cross to eliminate strain biases.

For promoter analysis, the 500 bp upstream of the transcriptional start site was taken for each Class-3 and known imprinted gene. Methylation levels for fetal and male six week frontal cortex (*Lister et al., 2013*; *Song et al., 2013*) and Histone H3 lysine 27 trimethylation (E16 and P0 forebrain) (*Gorkin et al., 2017*; *Shen et al., 2012*; *Sloan et al., 2016*) were extracted using UCSC Table Browser (*Karolchik et al., 2004*). The mean level across the 500 bp interval was calculated then the heatmap was produced using ggplot2 (*Wickham, 2016*).

## Allele-specific expression analysis

Raw sequencing read FASTQ files were downloaded from EMBL-EBI European Nucleotide Archive for each of the RNA-seq datasets (*Babak et al., 2015*; *Bonthuis et al., 2015*; *Crowley et al., 2015*; *Perez et al., 2015*; *Andergassen et al., 2017*). Low-quality bases and adapters were removed with trim_galore (v0.4.1) (Babraham Bioinformatics - Trim Galore!). SNPSplit (v0.3.4) (*Krueger and Andrews, 2016*) was used to separate reads by the parent of origin, which first required the preparation of allele-specific reference genomes for C57BL6/CAST_Eij and CAST_Eij/FVB (based on C57BL6) with the following commands SNPsplit_genome_preparation `--vcf_file` mgp.v5.merged.snps_all. dbSNP142.vcf.gz `--reference_genome` GRCm38_fasta/ `--strain  CAST_EiJ` and SNPsplit_ genome_preparation `--vcf_file` mgp.v5.merged.snps_all.dbSNP142.vcf.gz `--reference_ genome` GRCm38_fasta/ `--strain` CAST_EiJ `--strain2` FVB_NJ `--dual_hybrid`. VCF files for strain-specific SNPs were obtained from http://www.sanger.ac.uk/data/mouse-genomes-project.

The Clusterflow pipeline tool was used to enable running multiple jobs in parallel across multiple processors on an HPC, however, all scripts are also compatible with running on a single processor (*Ewels et al., 2016*). Trimmed reads were aligned to either the C57BL6/CAST_Eij or CAST_Eij/FVB reference genomes using HiSat2 (v2.1.0) (*Kim et al., 2019*), run via the hisat2 ClusterFlow module. Aligned reads were then name sorted to be compatible with SNPSplit, and run via the samtools (v1.9) (*Li et al., 2009*) Clusterflow module. Aligned files were run through SNPSplit to produce separated parent-specific alignment files using the SNP files produced by the genome preparation: all_ SNPs_CAST_EiJ_GRCm38.txt.gz    or    all_FVB_NJ_SNPs_CAST_EiJ_reference.based_on_GRCm38. txt.gz. A custom Clusterflow module was created for SNPsplit [SNPSplit.cfmod]. Gene counts from each parent-specific alignment BAM file produced by SNPSplit were calculated using featureCounts (v1.5.0-p2) (*Liao et al., 2014*) via Clusterflow. A custom Rscript DESeq2_featureCounts_2_CountsTables.R is used to make a single counts table, including normalized reads, from the individual featureCount files. All scripts to reproduce the analysis are freely available at https://github.com/darogan/ ASE_Meta_Analysis (copy archived at *Edwards et al., 2023*) For gene expression comparisons, the sum of the raw read counts per gene for both genomes was calculated. These values were converted to TPM (transcripts per million) then the mean value across biological replicates was calculated.

The counts tables were then manipulated into the configuration needed for the ISoLDE R package (*Reynès et al., 2020*). ISoLDE was used to test both allele-specific parental and strain biases. The default resampling method was used with nboot = 3000 for datasets with more than two replicates. For datasets with two replicates or fewer the threshold method was used (*Reynès et al., 2020*). Overlaps were then identified between the different datasets.

## Mice

All animal procedures were subject to local institutional ethical approval and performed under a UK Government Home Office license (project license number: PC213320E). Reciprocal crosses of

the mouse lines C57BL/6 J and castaneus (CAST/EiJ) were generated. For loss of IG-DMR studies, mutant mice were maintained on a C57BL6/J background by crossing males heterozygous for the deletion with wild-type females. For expression analysis, female or male heterozygotes were mated to castaneus (CAST/EiJ) mice to generate IG-DMR deletion heterozygotic conceptuses and wildtype littermates.

## Tissue collection

Samples were harvested from three to four mice from each cross at three developmental stages: e16.5 (16.5 days after conception), P7 (7 days after birth), and P60 (60 days after birth). At e16.5, the whole brain, liver, and placenta were harvested. At P7 and P60, the brain stem, cerebellum, cortex, hippocampus, and hypothalamus were harvested. Samples were snap-frozen in liquid nitrogen and stored at –80 degrees. For loss of IG-DMR studies samples were collected at E15.5, n=6 maternal heterozygotes and five wildtype littermates.

## qRT–PCR

DNA and RNA were extracted from samples using AllPrep DNA/RNA Mini kit (Qiagen) according to the manufacturer's instructions. 5 µg RNA was treated with DNaseI (Thermo Scientific) as per the manufacturer's instructions. 1 µg RNA was reverse transcribed with Revert Aid RT (Thermo Scientific). Assays were designed using PyroMark Assay Design SW 2.0 (Qiagen) and provided in *Supplementary file 1j*. The annealing temperature for each primer set was optimized by gradient PCR. The qPCR reactions were run on a LightCycler 480 (Roche) with Brilliant III Ultra-Fast SYBR Green qPCR Master Mix and the following conditions: 95 °C for 5 min followed by 45 cycles of 95 °C - 10 s, specific annealing temperature - 10 s, 72 °C - 10 s. All reactions were run in duplicate, and the relative expression of each gene was calculated using the ΔCt method and normalized to the housekeeping gene *Tbp*. Genes with expression lower than 0.05 times that of the housekeeping gene *Tbp* after qRT-PCR were not analyzed further as weak expression leads to inconsistent results between technical replicates in pyrosequencing.

## Allelic expression analysis

Parental allelic expression quantification was performed by pyrosequencing. Streptavidin Sepharose High-Performance beads (GE healthcare) dissolved in binding buffer (10 mM Tris-HCL pH7.6, 2 M NaCl, 1 mM EDTA, 0.1% Tween-20) were shaken with the qPCR product at 1400 rpm for 20 min. The biotinylated strand was purified using a PyroMark Q96 Vacuum Workstation (QIAGEN) then sequencing primers annealed in annealing buffer (20 mM Tris-acetate pH7.6, 2 M magnesium acetate) at 85 °C for 3 min. Sequencing was performed on a PyroMark Q96 MD pyrosequencer (Qiagen) using PyroMark Gold Q96 Reagents (Qiagen). The mean expression bias from three or four biological replicates of tissue was then calculated. Genomic DNA was also assessed to identify any amplification bias from the primers and all assays were corrected for this bias.

## Clonal methylation analysis

DNA (0.5–1 µg) was bisulfite treated using the two-step protocol of the Imprint DNA Modification Kit (Sigma). converted DNA was amplified using primers Fwd 5' TTGATGGAGTAAAAGGAATTGTTT TAGG and Rev 5' CCAATTCAAAAATTTAAAAAAAACAAAACC with HotStarTaq DNA Polymerase (QIAGEN). The PCR conditions were: (1) 95 °C – 5 min; (2) 94 °C – 30 s, 55 °C – 30 s, 72 °C – 55 s, 40 cycles; (3) 72 °C – 5 min. PCR Products were run on a 1.5% agarose gel, bands were then cut out and DNA was extracted using MinElute Gel Extraction Kit (QIAGEN). Purified DNA was ligated into the pGEM-T Easy Vector and transformed into Stellar Competent Cells (Cat# 636766). Selected colonies were Sanger sequenced by GENEWIZ. At least 20 clones were tested for each tissue and genotype from two biological replicates. Clones were assessed using BISMA - Bisulfite sequencing DNA Methylation Analysis (*Rohde et al., 2010*).

## Additional information

### Funding

| Funder | Grant reference number | Author |
|---|---|---|
| Medical Research Council | MR/R009791/1 | Anne C Ferguson-Smith |

The funders had no role in study design, data collection and interpretation, or the decision to submit the work for publication.

### Author contributions

Carol A Edwards, Conceptualization, Formal analysis, Investigation, Methodology, Writing – original draft, Writing – review and editing; William MD Watkinson, Stephanie B Telerman, Formal analysis, Validation, Investigation, Writing – review and editing; Lisa C Hulsmann, Conceptualization, Formal analysis; Russell S Hamilton, Formal analysis; Anne C Ferguson-Smith, Conceptualization, Supervision, Funding acquisition, Writing – review and editing

### Author ORCIDs

Carol A Edwards http://orcid.org/0000-0002-1887-1280
Russell S Hamilton http://orcid.org/0000-0002-0598-3793
Anne C Ferguson-Smith http://orcid.org/0000-0002-7608-5894

### Ethics

All animal procedures were subject to local institutional ethical approval and performed under a UK Government Home Office license (project license number: PC213320E).

### Decision letter and Author response

Decision letter https://doi.org/10.7554/eLife.83364.sa1
Author response https://doi.org/10.7554/eLife.83364.sa2

---

## Additional files

### Supplementary files

• Supplementary file 1. Data generated in this study. (a) Study information for Dataset A (*Babak et al., 2008*), Dataset B (*Bonthuis et al., 2015*), Dataset C (*Perez et al., 2015*), Dataset D (*Crowley et al., 2015*) and Dataset E (*Andergassen et al., 2017*). (b) All genes called in original studies. (c) Overlapping Novel genes called in original studies. (d) All genes called in this study using ISoLDE. (e) Number of genes called in individual tissues in the original study and this study. (f) Expression levels of genes in RNA-seq data used for ISoLDE and called as biased in original study - *Figure 2—figure supplement 1*. (g) Strain biased genes called in Dataset B and Dataset C in this study. (h) CpG Methylation and H3K27me3 over promoter regions of imprinted and Class-3 genes - *Figure 3D*. (i) Overlapping genes called in this study and the original one. This list includes genes generated in the undetermined list by the ISoLDE pipeline. (j) List of primers used in the study.

• MDAR checklist

### Data availability

All data generated or analysed during this study are included in the manuscript and supporting files. Allele specific pyrosequencing data and clonal bisulfite sequencing data generated in this study is available at https://doi.org/10.17863/CAM.90155.

The following dataset was generated:

| Author(s) | Year | Dataset title | Dataset URL | Database and Identifier |
|---|---|---|---|---|
| Edwards C, Watkinson W, Telerman S, Hulsmann L, Hamilton R, Ferguson-Smith A | 2023 | Research data supporting "Reassessment of weak parent-of-origin expression bias shows it rarely exists outside of known imprinted regions" | https://doi.org/10.17863/CAM.90155 | Apollo, 10.17863/CAM.90155 |

The following previously published datasets were used:

| Author(s) | Year | Dataset title | Dataset URL | Database and Identifier |
|---|---|---|---|---|
| Babak T, DeVeale B, Tsang EK, Zhou Y, Li X, Smith KS, Kukurba KR, Zhang R, van der Kooy D, Montgomery SB, Fraser HB, Jb Li | 2015 | Genetic conflict reflected in tissue-specific maps of genomic imprinting in human and mouse | https://www.ncbi.nlm.nih.gov/sra/?term=SRP020526 | NCBI Sequence Read Archive, SRP020526 |
| Bonthuis PJ, Huang WC, Stacher Hörndli CN, Ferris E, Cheng T, Gregg C | 2015 | Noncanonical Genomic Imprinting Effects in Offspring | https://www.ncbi.nlm.nih.gov/geo/query/acc.cgi?acc=GSE70484 | NCBI Gene Expression Omnibus, GSE70484 |
| Crowley JJ, Zhabotynsky V, Sun W, Huang S, Pakatci IK, Kim Y, Wang JR, Morgan AP, Calaway JD, Aylor DL, Yun Z, Bell TA, Buus RJ, Calaway ME, Didion JP, Gooch TJ, Hansen SD, Robinson NN, Shaw GD, Spence JS, Quackenbush CR, Barrick CJ, Nonneman RJ, Kim K, Xenakis J, Xie Y, Valdar W, Lenarcic AB, Wang W, Welsh CE, Zhang Z, Holt J, Guo Z, Threadgill DW, Tarantino LM, Miller DR, Zou F, McMillan L, Sullivan PF, Pardo-Manuel de Villena F, Cp Fu | 2015 | Analyses of allele-specific gene expression in highly divergent mouse crosses identifies pervasive allelic imbalance | https://www.ncbi.nlm.nih.gov/sra/?term=SRP056236 | NCBI Sequence Read Archive, SRP056236 |
| Perez JD, Rubinstein ND, Fernandez DE, Santoro SW, Needleman LA, Ho-Shing O, Choi JJ, Zirlinger M, Chen SK, Liu JS, Dulac C | 2015 | Quantitative and functional interrogation of parent-of-origin allelic expression biases in the brain | https://www.ncbi.nlm.nih.gov/geo/query/acc.cgi?acc=GSE67556 | NCBI Gene Expression Omnibus, GSE67556 |
| Andergassen D, Dotter CP, Wenzel D, Sigl V, Bammer PC, Muckenhuber M, Mayer D, Kulinski TM, Theussl HC, Penninger JM, Bock C, Barlow DP, Pauler FM, Hudson QJ | 2017 | Mapping the mouse Allelome reveals tissue-specific regulation of allelic expression | https://www.ncbi.nlm.nih.gov/geo/query/acc.cgi?acc=GSE75957 | NCBI Gene Expression Omnibus, GSE75957 |

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
