## [Editor Report]

This manuscript presents a useful meta-analysis of genes with parent-specific expression from mouse-published RNA-seq datasets, focusing on genes with weak allelic bias. A combination of systematic bioinformatic analysis and experimental validation convincingly shows that the number of parentally biased genes has been overestimated and the few novel ones lie at the periphery of known imprinted loci. The work will be of interest to genomicists with an interest in imprinting and its mechanisms.

---

## [Decision Letter]

**Decision letter after peer review:**

Thank you for submitting your article "Weak parent-of-origin expression bias: is this imprinting?" for consideration by *eLife*. Your article has been reviewed by 3 peer reviewers, and the evaluation has been overseen by a Reviewing Editor and Detlef Weigel as the Senior Editor. The reviewers have opted to remain anonymous.

Essential revisions:

The three reviewers overall appreciated the work, and the clarification it provides about the genome-wide occurrence of genomic imprinting. In addition, please provide a point-to-point answer to the reviewers.

1) Improve Figure 3 D.

2) Address the question as to why allelic ratios in the pyrosequencing analyses were not systematically normalized by amplification biases of gDNA.

3) Provide expression level comparison between true imprinted genes (confirmed ones) and the ones that were falsely called with (weak) parental bias.

*Reviewer #2 (Recommendations for the authors):*

Pcdhb12 – "This is not imprinting per se as does not reflect a bias in single cells but rather a population bias within the brain of individuals." This sentence should either have a reference to previous work about random monoallelic expression (if it is convincingly shown there) or it should be softened, since the data of this study doesn't allow making conclusions about population bias.

gDNA amplification bias in the pyrosequencing data – as indicated above, I don't understand (a) why this is not corrected for all loci. Even if the bias is very small, this would feel like the right thing to do, (b) if/when it's corrected, showing the 50-50 ratio in plots is meaningless and I'd suggest omitting it, or showing the original bias.

*Reviewer #3 (Recommendations for the authors):*

It appears that thresholding is used in calling allelic bias: "Expression was called as biallelic if the mean of the paternal expression from both the C57BL/6 x CastEiJ and CastEiJ x C57BL/6 crosses was between 45 and 55%". At the same time they provide the explanation of an important statistical flaw of such approaches: "expression levels can influence ASE calling: the lack of read depth in lowly expressed genes may erroneously lead to genes being called as biased, because a small difference in read numbers produces larger bias in weakly expressed transcripts". An additional source of FP is overdispersion (i.e. additional variation present compared to statistical model), which might introduce significant changes in width of confidence intervals.

---

## [Author Response]

1) Improve Figure 3 D.

We agree this is difficult to read. We have removed the names from the figure and improved the resolution of writing on axes and key. We have also provided a supplemental table with the corresponding gene names and methylation/H3K27 levels from which the figure was generated.

2) Address the question as to why allelic ratios in the pyrosequencing analyses were not systematically normalized by amplification biases of gDNA.

The PCR amplification bias has now been corrected for in every assay, and the corrected gDNA data now has been removed from each figure.

3) Provide expression level comparison between true imprinted genes (confirmed ones) and the ones that were falsely called with (weak) parental bias.

We have now taken the RNAseq data that was processed through the ISoLDE pipeline and extracted the transcripts per million (TPM) expression levels for each of the genes called in the original studies. We find no over representation of lowly expressed genes in the novel biased genes compared with known imprinted genes. This is now provided in Figure 2 —figure supplement 1. We also looked specifically at the expression levels of the genes tested by pyrosequencing in these datasets and saw no relationship between validation and expression levels. Expression levels are consistent between studies, especially in the same tissue, indicating the lack of reproducibility between studies is not due to differing expression. This has now been presented in the manuscript.

Reviewer #2 (Recommendations for the authors):Pcdhb12 – "This is not imprinting per se as does not reflect a bias in single cells but rather a population bias within the brain of individuals." This sentence should either have a reference to previous work about random monoallelic expression (if it is convincingly shown there) or it should be softened, since the data of this study doesn't allow making conclusions about population bias.

This sentence has now been rewritten “This suggests that the *Pcdhb12* bias does not reflect a maternal bias in each cell but rather a population bias within the brain of individuals: single cell analysis of neurons is needed to confirm this hypothesis.”

gDNA amplification bias in the pyrosequencing data – as indicated above, I don't understand (a) why this is not corrected for all loci. Even if the bias is very small, this would feel like the right thing to do, (b) if/when it's corrected, showing the 50-50 ratio in plots is meaningless and I'd suggest omitting it, or showing the original bias.

Thank you for this perspective. In response to the reviewer, the PCR amplification bias has now been corrected for all assays and the corrected gDNA data removed from each figure.

Reviewer #3 (Recommendations for the authors):It appears that thresholding is used in calling allelic bias: "Expression was called as biallelic if the mean of the paternal expression from both the C57BL/6 x CastEiJ and CastEiJ x C57BL/6 crosses was between 45 and 55%". At the same time they provide the explanation of an important statistical flaw of such approaches: "expression levels can influence ASE calling: the lack of read depth in lowly expressed genes may erroneously lead to genes being called as biased, because a small difference in read numbers produces larger bias in weakly expressed transcripts". An additional source of FP is overdispersion (i.e. additional variation present compared to statistical model), which might introduce significant changes in width of confidence intervals.

We used the threshold approach only to call bias in pyrosequencing data. This allows for the error in the pyrosequencing methodology which could lead to biallelic genes being called as biased. As stated in the methods we omitted genes that had low relative expression compared with TBP (<0.05) as we found that this led to large variation between technical replicates in the pyrosequencing.